# Single-cell transcriptomics unveil profiles and interplay of immune subsets in rare autoimmune childhood Sjögren's disease
Myung-Chul Kim[1,2,3,4,5], Umasankar De [1,2], Nicholas Borcherding[6], Lei Wang[1,2], Joon Paek[5,6], Indraneel Bhattacharyya[5,7], Qing Yu[8], Ryan Kolb[1,2], Theodore Drashansky[9], Akaluck Thatayatikom[10,11] ✉, Weizhou Zhang [1,2,11] ✉ & Seunghee Cha [5,7,11] ✉

Childhood Sjögren's disease represents critically unmet medical needs due to a complete lack of immunological and molecular characterizations. This study presents key immune cell subsets and their interactions in the periphery in childhood Sjögren's disease. Here we show that single-cell RNA sequencing identifies the subsets of IFN gene-enriched monocytes, *CD4*[+] T effector memory, and *XCL1*[+] NK cells as potential key players in childhood Sjögren's disease, and especially in those with recurrent parotitis, which is the chief symptom prompting clinical visits from young children. A unique cluster of monocytes with type I and II IFN-related genes is identified in childhood Sjögren's disease, compared to the age-matched control. In vitro regulatory T cell functional assay demonstrates intact functionality in childhood Sjögren's disease in contrast to reduced suppression in adult Sjögren's disease. Mapping this transcriptomic landscape and interplay of immune cell subsets will expedite the understanding of childhood Sjögren's disease pathogenesis and set the foundation for precision medicine.

Sjögren's disease in adults (aSjD) occurs mainly in peri-menopausal female patients with clinical hallmarks of dry mouth and dry eyes (sicca symptoms)[1]. Of those, 30–40% of patients present extra glandular manifestations in the lungs, kidneys, stomach, skin, and/or neural system[2]. The estimated interval between the initial presentation of clinical symptoms and diagnosis is 6–10 years[3]. Rapid advances in comprehensive transcriptomic profiling at the single-cell level in recent years have identified potential biomarkers, genes, and pathways that may contribute to aSjD pathogenesis[4,5].

Accumulating evidence supports the increasing incidences of SjD occurring in children (childhood SjD, cSjD)[6]. cSjD has long been considered as a rare autoimmune disease affecting children and adolescents younger than 20 years of age[7] with an estimated prevalence of 1% of SjD patients[8]. However, since there is no established diagnostic criterion for cSjD, the exact

prevalence of cSjD can be underestimated. Currently, cSjD diagnosis is based on the 2016 American College of Rheumatology and the European League Against Rheumatism (ACR/EULAR) for primary aSjD[7] or experts' opinions. Previous case reports indicate that most children with cSjD lack subjective sicca symptoms[7,9–11], although our recent study showed that the number of cSjD patients with objective secretory dysfunction was significantly higher than in non-cSjD patients (40.7% vs. 12.8%, p < 0.004)[10], prompting a need for close evaluation of cSjD patients for their secretory function, regardless of the absence of patient-reported complaint of dryness.

One of the notable clinical characteristics of cSjD is recurrent parotitis (RP), which tends to precede oral and ocular symptoms[9,10,12]. The cause of frequent RP, known to serve as one of the predictive factors for lymphoma development in 5% of aSjD patients, in the pathogenesis of cSjD are completely unknown. To our best knowledge, the University of Florida (UF)

[1]Department of Pathology, Immunology and Laboratory Medicine, University of Florida College of Medicine, Gainesville, FL 32610, USA. [2]UF Health Cancer Center, University of Florida, Gainesville, FL 32610, USA. [3]Diagnostic Laboratory Medicine, College of Veterinary Medicine, Jeju National University, Jeju 63243, Republic of Korea. [4]Research Institute of Veterinary Medicine, College of Veterinary Medicine, Jeju National University, Jeju 63243, Republic of Korea. [5]Center for Orphaned Autoimmune Disorders, University of Florida College of Dentistry, Gainesville, FL 32610, USA. [6]Department of Pathology & Immunology, Washington University School of Medicine in St. Louis, St Louis, MO 63110, USA. [7]Department of Oral & Maxillofacial Diagnostic Sciences, University of Florida College of Dentistry, Gainesville, FL 32610, USA. [8]The Forsyth Institute, Cambridge, MA 02142, USA. [9]Cellularity, Inc. 170 Park Ave, Florham Park, NJ 07932, USA. [10]AdventHealth Medical Group, Orlando, FL 32643, USA. [11]These authors contributed equally: Akaluck Thatayatikom, Weizhou Zhang, Seunghee Cha.
✉e-mail: akaluck1@hotmail.com; zhangw@ufl.edu; scha@dental.ufl.edu

cSjD cohort, established by a multi-disciplinary team of experts in 2018, is the only one prospective cohort in the nation with the banking of biospecimens and comprehensive clinical/laboratory data. In this cohort, the prevalence of anti-Ro/SSA and RP was 56% and 70% calculated in early 2021[10], respectively. This may suggest more diversified antigens and antigenic epitopes present in cSjD salivary glands compared to aSjD's and imply potential usefulness of RP in diagnosing cSjD.

Single-cell RNA-sequencing (scRNA-seq) has become an essential tool in SjD research, which allows the establishment of transcriptomic profiles at a single-cell level in an unbiased manner[13,14]. To address unmet medical needs in cSjD, we performed scRNA-seq on four PBMC samples per group purified from the groups of 1) cSjD (two with RP and two without RP), 2) non-cSjD (symptomatic patients who failed to meet the 2016 SjD criteria), 3) Bx (non-cSjD with biopsy positivity), 4) BxRP (non-cSjD with biopsy and RP positivity), and 5) HC (healthy controls). Our comprehensive transcriptomic analysis provides insights into key immune cell players and targetable genes/pathways that distinguish cSjD from non-cSjD or aSjD, contribute to biopsy positivity, and drive RP.

## Methods

### Study subjects for PBMC collection and single-cell isolation

Pediatric male and female participants who were referred from the Pediatric Rheumatology have been enrolled at the Center for Orphaned Autoimmune Disorders (COAD), UF Health/Shands, Gainesville, Florida. All subjects provided written informed consent following the study protocol approved by the UF Institutional Review Board and the study was conducted in accordance with the Declaration of Helsinki.

PBMCs were prepared from fresh blood samples by Ficoll-Paque PLUS (GE Healthcare, Chicago, IL) density gradient centrifugation. SepMate-50 tube (Stem Cell Technologies, Vancouver, Canada) was used to isolate PBMCs, according to the manufacturer's instructions. PBMCs were washed with Dulbecco's phosphate-buffered saline (PBS) (Thermo Fisher Scientific, Waltham, MA) supplemented with 2% heat-inactivated fetal bovine serum (FBS) (Life Technologies, Pleasanton, CA). PBMCs were cryo-preserved in liquid nitrogen until analysis.

The current study compared five groups with four PBMC samples in each group: HC, cSjD, Non-cSjD, Bx, and BxRP (Supplementary Table 1). The diagnosis of cSjD was made according to the 2016 ACR/EULAR criteria[15] for primary SjD and the age range of cSjD was from 10 to 17. Two cSjD patients exhibiting RP were included in cSjD. Those who did not fulfill the 2016 criteria were categorized as non-cSjD (ages ranging from 12 to 24). Interestingly, some non-cSjD patients presented positive lip biopsies without fulfilling the 2016 criteria, thus being classified as Bx (ages ranging from 8 to 18). Some of these biopsy-positive non-SjD patients presented RP, thus being categorized as BxRP (ages ranging from 7 to 19). Two subjects were older than 18 years old, more specifically 24 and 19, but included in this study as their disease onset was seven and eight years ago, respectively, thus still being seen by a pediatric rheumatologist. Young adolescents or young adults were recruited as health controls (HCs) (ages ranging from 18 to 31) since healthy children under 18 rarely visit rheumatology clinics in general.

The summary of the demographic characteristics of the subjects for this study (Supplementary Fig. 1) and the general information on the UF cohort[10] are available. The majority of subjects presented fatigue, arthralgia, and/or sicca symptoms regardless of their final diagnosis, and they happened to be anti-SSA-negative except for one patient in this study. A total of six out of 16 exhibited or reported a history of RP.

### Cell hashing and fluorescence-activated cell sorting

Following the purification and single-cell isolation of PBMC, the cells were incubated with an Fc receptor-blocking reagent (Miltenyi Biotec, Germany) for 10 min on ice. Subsequently, cells were stained with an oligo-tagged TotalSeq Type B hashtag antibody at a final concentration of 4 µg/ml for 30 min on ice. After washing, cells were incubated with anti-human CD45 FITC (clone HI30, BioLegend, USA) and Fixable Viability Dye eFluor780 (eBioscience, USA) for 30 min on ice. Live single CD45$^+$ cells were sorted using FACSAria III (BD Biosciences Pharmingen, USA) and subject to loading onto the Chromium controller.

### 10 × Genomics single-cell RNA library generation, processing, and data preparation

Standardized single-cell library preparation was performed per 10 × Genomics Next GEM Single Cell 3' v3.1 GEM and Single Cell 3' Feature Barcode Library Kit (10 × Genomics, USA). Libraries were sequenced using Illumina NovaSeq, S4, 2 × 150 (750 Gb). Demultiplexing of the Illumina files and generation of FASTQ files containing the scRNA-seq data were performed using the cellranger mkfastq function in the Cell Ranger software package (v. 3.1.0). Alignment of scRNA-seq reads to the human reference genome (GRCh38) and transcript quantification was performed using the cellranger *count* function. R toolkit Seurat (v. 4.0.6) was used for the data processing, generating the Seurat object as an input file on RStudio (v. 4.2) for subsequent bioinformatic processes.

### scRNA-seq data integration, initial pre-processing, and sub-clustering

Seurat objects of all twenty samples were merged and integrated into one master object[16,17]. Low-quality of cells with either unique feature counts of less than 200 or over 4000 or mitochondrial counts of more than 10% were filtered out. Samples were normalized with the default setting. Preparation for integration used 3,000 anchor features. Principal component analysis (PCA) was used for linear dimensional reduction. The dimensionality of the dataset was determined as 30, which was based on the most significant principal component (PC) ($P < 1e-5$) from the Jackstraw substitution test algorithm. t-distributed stochastic neighbor embedding (t-SNE) was used for graph-based clustering with a resolution of 1.2. Immune cells were identified and labeled by using SingleR (v. 1.8.1)-based unbiased cell type recognition[18], canonical markers for lineage, or markers for rare and unique populations from previous publications[14]. The Celldex package (v. 1.6.0) was used to leverage reference signatures of pure cell types to infer the cell of origin of every single cell. To scrutinize functionally unique immune subsets, major immune cell populations were sub-clustered and subject to singlet selection. scDblFinder (v. 1.4.0) R package was used to remove potential doublets[16]. Doublets or clusters that were not assigned to all groups due to very low numbers were excluded from sub-cluster-based analyses. For the scRNA-seq data comparison between cSjD and adult SjD in PBMCs[14], we used processed scRNA-seq data deposited in Gene Expression Omnibus (GEO) Datasets (GSE157278) and applied the same procedures of data preparation as described in this section.

### The study workflow and the quality control of the data

The library and cell multiplexing statistics (Supplementary Data 19) and the standard pre-processing and quality control of the data (Supplementary Fig. 2a–c) are available online. A high quality of 53,282 cells were obtained from HC ($n = 11,288$), cSjD ($n = 10,590$), Bx ($n = 12,860$), BxRP ($n = 9716$), and non-cSjD ($n = 8828$) groups (Supplementary Fig. 2d). Immune cells across the groups shared a similar global structure of RNA expression on the two-dimensional plots (Supplementary Fig. 2d, e). Bx, cSjD, and non-cSjD patients showed a significant alteration in the number of genes detected in a cell compared to HC (Supplementary Fig. 2f) while presenting similar cell numbers across the groups (Supplementary Fig. 2g). The number of genes per cell was normalized by applying sequencing depth to each sample, which yielded a slightly decreased gene number per cell in non-cSjD compared to HC (Supplementary Fig. 2h).

### Identification of the major immune cell subsets by canonical markers

Using the 10x Genomics platform, we generated a total of five droplet-based scRNA-seq libraries (Supplementary Fig. 3a). Among the total of 29 clusters identified in the integrated Seurat dataset (Supplementary Fig. 3b), we first assessed clustering performance by identifying the main immune cell types and found a clear separation of T, B, natural killer (NK),

and myeloid populations (Supplementary Fig. 3c). Out of the top 20 DEGs to define each major immune subset, representative genes are presented on the heatmap (Supplementary Fig. 3d). Functionally distinct immune cell subsets are also detected based on various canonical markers (Supplementary Fig. 3e).

Our clustering performance was further confirmed by unbiased cell type recognition (Supplementary Fig. 2i), following the exclusion of double-negative T cells that barely expressed *CD4* or *CD8A* (Cluster 17) and doublets (Clusters 23 and 26). Interestingly, the proportion (%) of *CD4+* T cells significantly decreased in cSjD compared to HC while the fraction of NK cells in cSjD and non-cSjD showed an increasing trend compared to HC (Supplementary Fig. 2j).

## DEG analysis
The likelihood-ratio test to find the differential expression for a single cluster, compared to all other cells, was used. To identify cluster markers and DEGs for all clusters across groups, the FindAllMarkers function was used in the Seurat package with the absolute $\log_2$-fold change threshold > 0.36 and $P$ value < 0.05[14]. For this study, DEGs were defined as differentially expressed genes detected by comparing any group of interest with HC unless otherwise specified. We defined cSjD-related DEGs as DEGs identified in the groups of Bx, BxRP, and cSjD for their lip biopsy positivity in common, while biopsy-related DEGs were derived from the comparison between cSjD-related DEGs and non-cSjD DEGs. In this particular set-up, cSjD-specific DEGs refer to cSjD DEGs compared to Bx and BxRP DEGs. In addition, we defined cSjD-specific Treg DEGs compared to adult SjD Treg clusters. Non-cSjD-specific DEGs refer to genes showing differential expression in non-cSjD compared to HC, Bx, BxRP, and cSjD. To avoid batch effects between cSjD and SjD scRNA-seq data, such as age or ethnicity difference, DEGs were obtained by comparing disease and its corresponding HC Treg clusters in the integrated scRNA-seq dataset.

## Gene set enrichment analysis (GSEA) and Gene Ontology analysis
Single-cell GSEA was performed using the escape R package (v. 1.6.0)[16]. Transformed z-scores are used to display GSEA results. Z-score represents the number of standard deviations a data point is from the mean as a standard score. Gene sets were derived from the Hallmark library of the Molecular Signature Database[19] and adopted from the previous publications[20–22]. DEGs were also subjected to either Gene Ontology Enrichment analysis using PANTHER annotation datasets (http://geneontology.org/) or ShinyGO (v. 0.741), a graphical gene-set enrichment tool[23]. The gene ontology results were filtered based on the criteria of $p$-value < 0.05 and false discovery rate (FDR) < 0.05. DittoSeq (v. 1.4.4) and pheatmap (v. 1.0.12) R packages were used to visualize gene sets defining specific molecular and biological pathways[24].

## Cell cycle analysis
Cell cycle assignment was performed by using the CellCycleScoring function and calling cc.genes.updated.2019 in Seurat[17,25]. Briefly, each cell was assigned a score based on its expression of G2/M and S phase markers, which were used for predicting the classification of each cell in the G2/M, S, and G1 phases.

## Cell-to-cell interaction analysis
CellChat R package (v. 1.4.0) was used to quantitatively infer intercellular communication networks from scRNA-seq data[26]. Doublets, *CD4-CD8-*, or *CD4+CD8+* T cells in integrated scRNA-seq dataset applied by the resolution of 1.7 were not analyzed. Multiple CellChat objects derived from each group were integrated into one master object for comparative analysis. The netVisual_heatmap function was used to visualize overall cell-cell communication. To find a potential ligand-receptor pair, the subsetCommunication function was used with the ligand $\log_2$-fold change threshold > 0.2, receptor log-fold change threshold > 0.1, and $P$ value < 0.05.

## Flow cytometric analysis
Cytek Aurora (Cytek, USA) and FlowJo software (TreeStar, USA) were used for flow cytometric data acquisition and analysis[17]. For the IFN stimulation experiment, PBMC was cultured in RPMI with 2-mercaptoethanol (Sigma, 55 µg/ml) and stimulated with 10 ng/ml of IFN-γ (R & D system) for 30 min Single cell suspension harvested was subjected to a 1% protease and phosphatase inhibitor cocktail (Sigma) for 1 hour, followed by surface and intracellular staining. A list of antibodies and relevant reagents used in this study is available (Supplementary Table 2).

## Treg suppression assay
For Treg suppression assay[17], naïve CD4+CD25- T cell and CD14+ monocytes were prepared using EasySep™ negative and positive selection kits (StemCell Technologies, Canada), respectively. Recombinant human GM-CSF and IL-4 (R & D system, USA) were used to differentiate monocytes into dendritic cells (DCs). FACS aria III (BD Biosciences, USA) was used to sort live CD45+CD3+CD4+CD25+CD127- Tregs from PBMCs. Carboxy fluorescein succinimidyl ester (CSFE) (Invitrogen, USA) was used to assess T cell proliferation. Anti-human CD3 antibody (clone: OKT3, BD PharMingen, USA) and autologous DCs were used to induce T cell proliferation. The results were normalized to live CD4+Foxp3-CSFE^low T cells cultured without Tregs and presented as % with the proliferating T cells set at 100%. In our study, a total of three pediatric cSjD patients (ages ranged from 12 to 18, newly diagnosed) samples and three adult SjD patients (ages ranged from ranging from 53 to 59, newly diagnosed or diagnosed for 1 or 5 years) samples with a total of six healthy controls (ages ranged from 15 to 54) were used for the Treg suppression assay.

## Statistics and reproducibility
For bioinformatic analysis, default statistical methods available within the Seurat package were used in this study. Non-parametric Wilcoxon rank-sum test was used to compare the significance of two-sample differential expression in the FindAllMarkers function. One-way analysis of variance (ANOVA) available within the ggpubr R package (v. 0.4.0) was used for statistical tests for the distribution of genes on count-level mRNA data. The two-sample significance testing utilized Welch's *T*-test, and the significance testing for more than three samples utilized one-way ANOVA. A total of six PBMC samples derived from three pediatric cSjD and three adult cSjD patients were subject to an IFN stimulation assay. A total of 12 PBMC samples derived from three pediatric cSjD, three adult SjD patients, and six healthy controls were subject to two independent Treg suppression assays. A total of 13 PBMC samples derived from six pediatric cSjD and six healthy controls were subject to two independent flow cytometric analyses to study the proportion of monocyte subsets.

## Reporting summary
Further information on research design is available in the Nature Portfolio Reporting Summary linked to this article.

# Results
## cSjD myeloid cell clusters exhibit gene signatures distinct from those in aSjD
Figure 1a presents the graphic summary of the 5 groups compared in this study. Considering current attention to the type I IFN signature in aSjD pathogenesis[27,28], we first focused on the six myeloid clusters identified (Fig. 1b) following sub-clustering and doublet exclusion of *CD68+* cells (Supplementary Data 1 and Supplementary Fig. 4a). Based on the differentially expressed genes (DEGs) analyzed across the clusters, canonical myeloid markers, and unbiased cell type annotation, we further defined each myeloid subset (Figs. 1c, d and Supplementary Fig. 4b). The *CD14+CD16+* intermediate monocytes were very few and not clearly separated as a subset. Interestingly, Cluster 4 was strongly characterized by DEGs associated with the type I IFN signature, including *IFIT1, IFI44L, IFI44, MX1,* and *ISG15,* along with *STAT1, MX2,* and *PNPT1* (Fig. 1c and e).

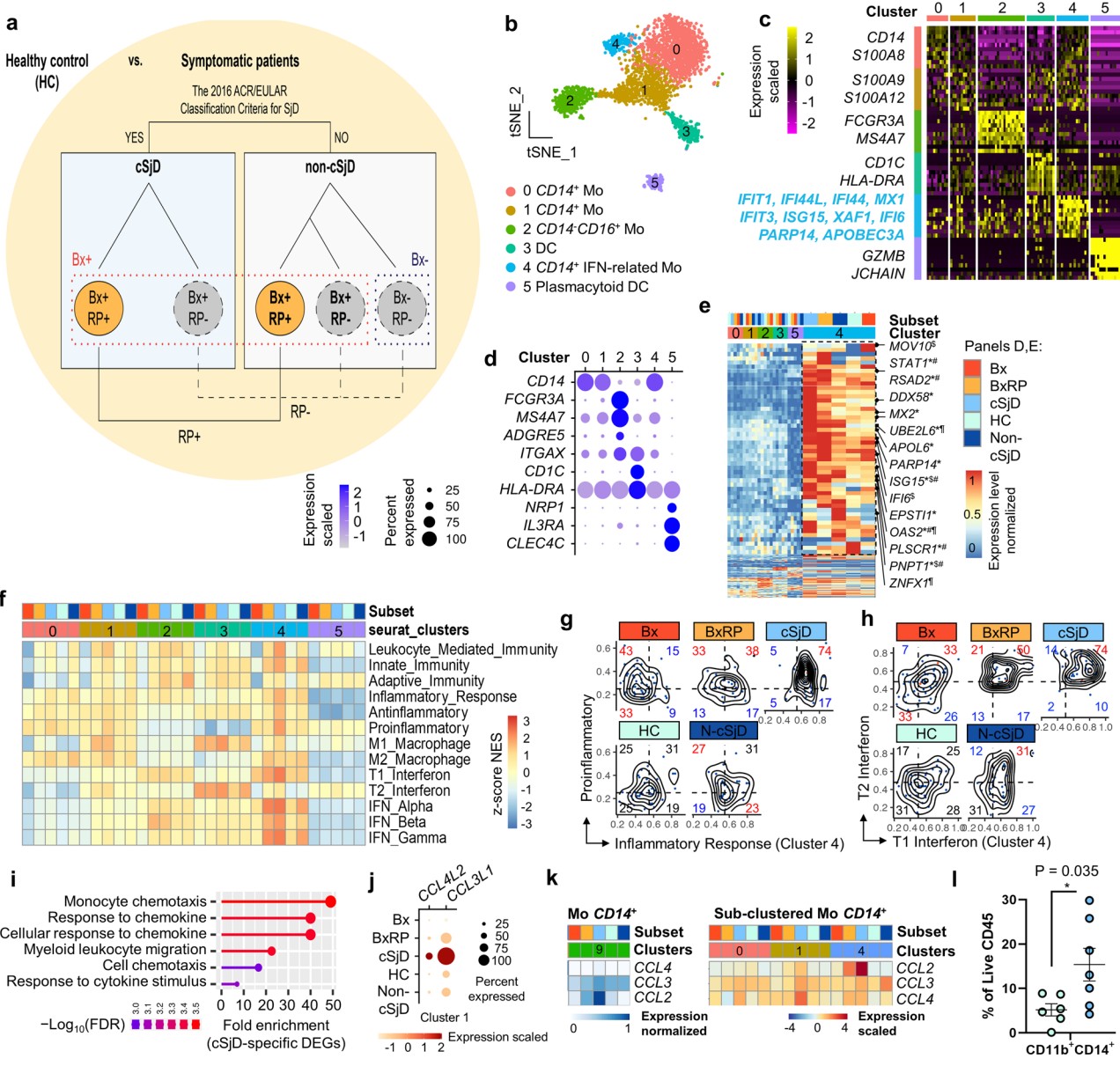

**Fig. 1 | Identification and characterization of myeloid subpopulations by scRNA-seq. a** The graphic summary of the five groups compared in this study. HC healthy control, cSjD childhood Sjögren's disease, RP recurrent parotitis, Bx biopsy, BxRP biopsy positivity with recurrent parotitis, Non-cSjD non-childhood Sjögren's disease; 2016 criteria, The 2016 ACR/EULAR criteria for aSjD. **b** Six myeloid subsets are presented on the tSNE plot. **c** Among the top 20 DEGs defining each subset, representative genes are presented on the heatmap, with Cluster 4 characteristically distinguished from others by the distinct upregulation of IFN-related genes. **d** Dot plot presents functional myeloid cell subset markers. **e** IFN-related genes are present on the heatmap, which is preferentially enriched and upregulated in cluster 4 of cSjD and, to a lesser extent, in BxRP, compared to HC. Genes listed on the heatmap refer to differentially upregulated genes with statistical significance between the group of interest and HC ($p < 0.05$, non-parametric Wilcoxon rank-sum t-test). *cSjD, #BxRP, $Bx, and ¶Non-cSjD, compared with HC. **f** GSEA reveals differentially enriched immune-associated signatures across the subsets and groups on the

heatmap. Hex density enrichment plot reveals an enrichment pattern of proinflammatory and inflammatory responses **g** and type I and II IFN signatures **h** in Cluster 4 of each group. Red and blue numbers on each quadrant illustrate upward and downward trends, respectively, compared to HC. **i** Gene ontology analysis using cSjD-specific myeloid DEGs reveals significant enrichment of biological processes associated with chemotaxis. **j** Dot plot reveals cSjD-specific upregulation of *CCL4L2* and *CCL3L1* in Cluster 1. **k** Expression pattern of CC chemokine ligands on the heatmap. Cluster 9 in the left Heatmap refers to $CD14^+$ monocytes in the original tSNE plot. **l** Flow cytometric analysis of PBMCs shows a significant expansion of circulating classic monocytes in cSjD, compared to HC. Error bar refers to standard deviation. One experiment for each child and adult group was performed. Statistical significance was obtained by comparing two groups of interest with a two-tailed unpaired Student's t-test. *p-value < 0.05. NES normalized enrichment score, HC healthy control, cSjD childhood Sjögren's disease, BX biopsy-positive non-cSjD without RP, BXRP biopsy-positive non-cSjD with RP.

We next performed gene set enrichment analysis (GSEA)[29,30] to compare the functional characteristics and biological states of myeloid subsets across the groups. GSEA revealed that in the majority of patients, except for the Bx group, myeloid cells showed a shift toward a more inflammatory phenotype compared to HC, evidenced by several immune-associated gene signatures, such as

leukocyte-mediated immunity, inflammatory response, and M1/2-like macrophage (Supplementary Fig. 4c, top). Among the symptomatic groups, BxRP and cSjD were skewed more towards the M1-like phenotype than M2-like, whereas non-cSjD contained two distinct populations of myeloid cells with either M1-like or M2-like phenotype (Supplementary Fig. 4c, bottom).

For Cluster 4, the enrichment pattern of gene sets in cSjD was strongly associated with inflammatory modulation (Fig. 1f, g) and the IFN-related pathways as predicted, followed by BxRP, compared to HC (Fig. 1h). DEGs, such as *ISG15*, *HERC5*, *UBE2L6*, and many MHC class II-related genes, from Cluster 4 between cSjD and HC were also significantly enriched with gene ontology terms of ISG15 protein conjugation and antigen presentation pathways (Supplementary Fig. 4h, i). In addition, other inflammatory cytokine pathways, such as TNF-α, TLR, IL-10, and IL-15, were also enriched in cSjD compared to HC (Supplementary Fig. 4d). Although Cluster 4 was identified in non-cSjD, gene sets in non-cSjD were different from Cluster 4 of cSjD. For example, IL-7 and IL-21 in Cluster 2, IL-21 in Cluster 4, and IL-4 and IL-21 in Cluster 5 were noted rather than the IFN signature. Interestingly, GSEA results did not support inflammatory roles of pDC (Cluster 5) in cSjD (Fig. 1f and Supplementary Fig. 4d). Dendritic cells (DCs) (Cluster 3) in BxRP were found to upregulate tryptophan metabolism. There was no identifiable enrichment pattern that was preferentially altered in Bx and BxRP compared to non-cSjD.

Furthermore, we identified 984 cSjD DEGs (Supplementary Data 2) with 40 cSjD-related (Supplementary Data 3), 29 cSjD-specific (Supplementary Data 4), and 2 Bx-specific myeloid DEGs (Supplementary Data 5 and Supplementary Fig. 4j) through multiple comparisons across the groups. Interestingly, cSjD-specific DEGs were significantly enriched with genes associated with chemotaxis and response to chemokines, such as *CCL4L2* and *CCL3L1*, especially in Cluster 1 of cSjD (Fig. 1i, j). Expression of chemokines, such as *CCL2*, *CCL3*, and *CCL4* in *CD14*+ monocytes (Cluster 9 in the original tSNE plot), exhibited an increasing tendency in cSjD relative to Bx and BxRP (Fig. 1k, left). The expression of *CCL2* was specific for IFN-related monocytes, and the significant upregulation was identified in cSjD as well as in BxRP, but not in non-cSjD (p = 0.0849 and $\log_2$-fold change = 0.3603) (Fig. 1k, right, and Supplementary Data 2).

We also analyzed DEGs that define non-cSjD, compared to HC, cSjD, and Biopsy+ (Bx combined with BxRP), and identified that *PLCG2* expression was specifically and highly upregulated in all non-cSjD myeloid subsets, compared to other groups (Supplementary Fig. 4e). Interestingly, *PLCG2* upregulation was found to be associated with the gene sets of IL-10, IL-23, and TNF-α, which was observed both in classic and non-classic monocytes, but not in IFN-related monocytes (Supplementary Fig. 4f). Our flow cytometric analysis revealed a significant increase in the proportion of circulating monocytes in cSjD compared to HC (Fig. 1l), whereas intermediate or non-classic monocytes remained unaltered (Supplementary Fig. 4g).

To compare cSjD monocytic clusters with those of aSjD, we independently analyzed the published scRNA-seq dataset of aSjD PBMC[14], which was the only accessible aSjD dataset currently. Following the same data integration, such as myeloid sub-clustering and doublet removal, 10 myeloid subsets were identified in aSjD (Supplementary Fig. 5a). However, we found that those myeloid clusters were neither enriched with genes associated with IFN pathways (Supplementary Fig. 5b and Supplementary Data 6) nor those of the inflammatory roles of myeloid cells in aSjD (Supplementary Fig. 5c). Consistently, a recent study, which analyzed the same aSjD scRNA-seq dataset, also failed to identify the IFN-related monocytic subpopulation in aSjD[31]. However, concluding whether it is due to cohort variability or true differences between cSjD and aSjD monocyte subsets will require further investigation. We further investigated if IFN-related monocytes are related to age or ethnicity of the subjects involved. With the same analysis applied, various age and ethnicity of healthy PBMC scRNA-seq datasets, including 4 groups of 11 children (GSE148633, GSE206295, GSE168732, and phs003048.v1.p1), 3 groups of 9 adults (GSE148633, GSE216489, and GSE211560), and a group of 2 centenarians (NBDC Human Database hum0229.v1), were subject to integrated analysis and sub-clustering of myeloid cells (Supplementary Fig. 5d). Interestingly, we identified IFN-related monocytes (Cluster 7) that were characterized by DEGs associated with the IFN signature, including *MX1*, *IFI44L*, *MX2*, *ISG15*, *IFIT3*, *XAF1*, *PARP14*,

and *IFIT2* (Supplementary Fig. 5e). This cluster was identified in PBMC samples of different ages (Supplementary Fig. 5f) and ethnicity groups (Supplementary Fig. 5g). GSEA clearly shows that IFN-related monocytes in healthy children were not enriched with proinflammatory and M1 macrophage signatures compared to adult monocytes (Supplementary Fig. 5h). Finally, we compared the enrichment scores of IFN-related gene signatures in IFN-related monocytes between cSjD and HC. Interestingly, the IFN-related monocytes from cSjD showed significantly higher enrichment scores of types I and type II IFN gene signatures, compared to HC, which was not associated with age or ethnicity (Supplementary Fig. 5i). Thus, IFN-related monocytes exist in both healthy child and adult individuals as well as cSjD monocytes. However, only the ones in cSjD are significantly enriched with IFN gene signatures.

### cSjD *CD4*+ T cells are characterized by activated effector memory *CD4*+ T cells and functionally intact Tregs

As the expansion of circulating cytotoxic *CD4*+ T cells has been reported in aSjD[14], we investigated the functionality of *CD4*+ T cells next. A total of nine *CD4*+ T cell clusters were identified by using unbiased cell type identification and the expression pattern of canonical/cluster markers (Figs. 2a–c and Supplementary Fig. 6a). Cluster 5 was classified as effector memory *CD4*+ T cells by upregulated granzyme genes, such as *GZMK* and *GZMA*, and memory cell markers, such as *CD27* and *CXCR4*, compared to effector-like *CD4*+ T cells (Cluster 4). In Cluster 5 from cSjD, gene sets associated with proinflammatory, cytolysis, and cytolytic granules were preferentially enriched (Supplementary Fig. 6b and Fig. 2d, top) and upregulated, which includes *EOMES*, *IL17A*, *IL1B*, and *TBX21* (Fig. 2d, bottom). Similarly, genes involved in cytotoxic function in effector memory *CD4*+ T cells[32], such as *SELL*, *GZMA*, *PRF1*, *NKG7*, and *GNLY*, and genes enriched with gene ontology term cytolysis were significantly upregulated in Cluster 5 of cSjD compared to that of HC (Fig. 2e, f). However, this effector memory subset was not enriched with Th1, Th2, and Th17 gene sets (Supplementary Fig. 6b), while Cluster 4 of effector-like *CD4*+ T cells was slightly enriched with the T helper responses in cSjD.

Tregs were identified in two clusters in which Cluster 8 showed a higher expression of genes associated with Treg functionality and MHC II molecules such as *HLA-DRB1* and *HLA-DPB1* than those in Cluster 7 (Fig. 2a–c). Additionally, many gene sets associated with Treg functionality, such as leukocyte-mediated immunity, glycogen metabolism, ROS, and TGF-β, were more enriched in Cluster 8 in cSjD patients than in HC (Supplementary Fig. 6b). Thymocyte selection associated high mobility group box (TOX), which is required for the canonical CD4 T cell lineage, NKT and Treg development, and involved in T cell exhaustion[33], was one of the most significantly upregulated genes in Cluster 8 Tregs from cSjD patients compared to all other groups (Fig. 2g, h).

As Tregs from aSjD is known to show functionally defective, we examined the functionality of Tregs from cSjD and HC by in vitro Treg suppression assay[34]. Interestingly, Tregs from cSjD patients were functionally intact, contrasting to aSjD Tregs, which were less suppressive compared to Tregs from HC (Fig. 2i and Supplementary Fig. 6c). Using sub-clustered *CD4*+ T subsets in the integrated cSjD and aSjD scRNA-seq datasets (Supplementary Fig. 6d), we determined DEGs that can mediate the suppressive capacity of cSjD Tregs (Supplementary Data 7). Interestingly, genes associated with inflammation, Treg functionality, and T cell exhaustion, such as *HLA-DQA2*, *JUNB*, *TOX*, and *CTLA4*, were upregulated in cSjD Treg clusters that were not identified in aSjD Tregs (Supplementary Data 7 and Fig. 2g). We identified 567 cSjD DEGs (Supplementary Data 8) with 13 cSjD-related (Supplementary Data 9), and 30 cSjD-specific DEGs for *CD4*+ T subsets (Supplementary Data 10).

Finally, to investigate T cell functionality in cSjD, CD4+ T cells were stimulated with IFNγ, resulting in a remarkable increase in phosphorylated STAT1 expression (269.2% increase, *p* < 0.01) in cSjD CD4+ T cells compared to HC (Fig. 2j). Of note, the magnitude of the increase was significantly higher in cSjD than in aSjD by ~8 times.

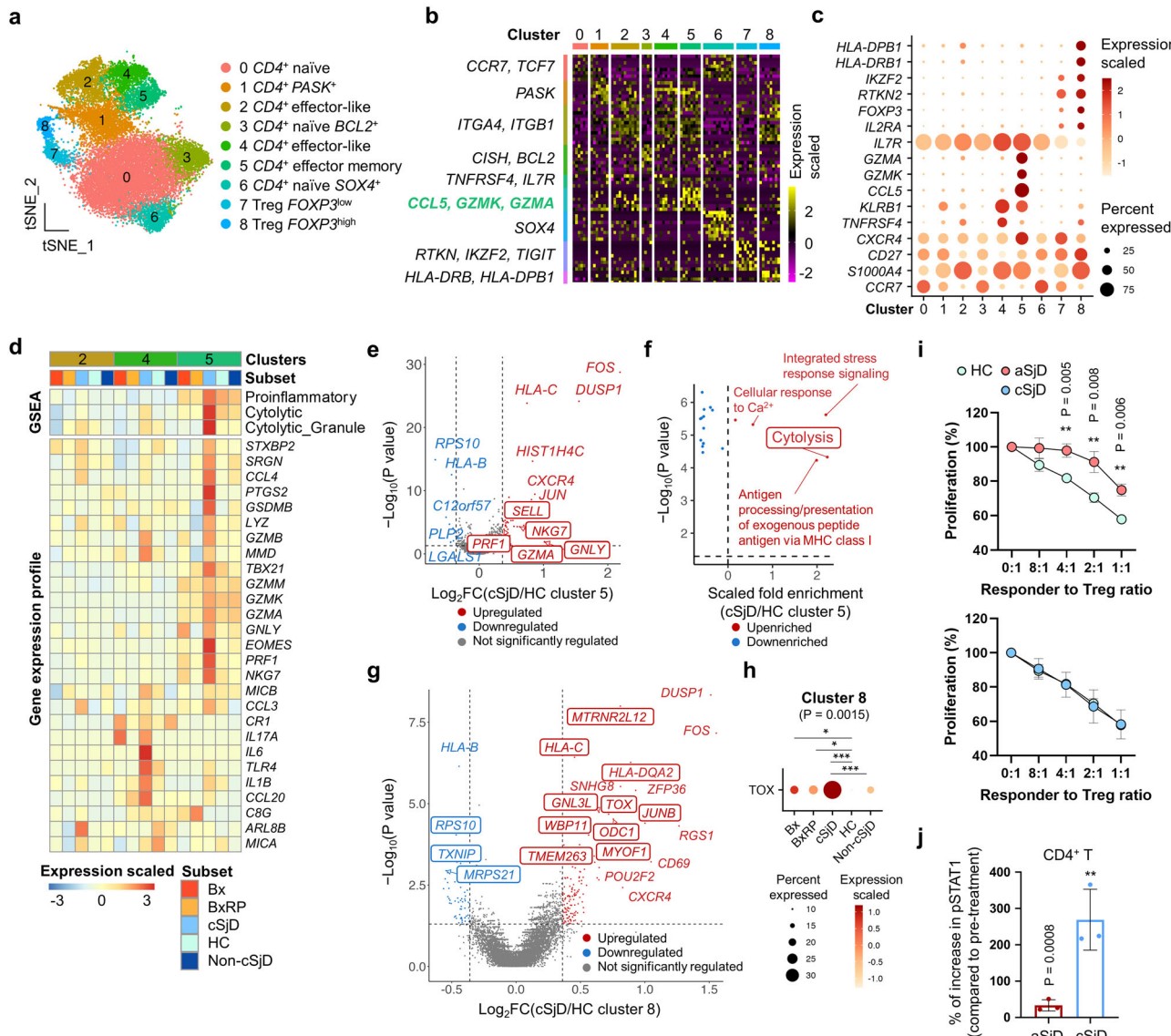

**Fig. 2 | Identification and characterization of *CD4*⁺ T cell subpopulations by scRNA-seq. a** A total of nine *CD4*⁺ T cell subsets are presented on the tSNE plot. **b** Among the top 20 DEGs in each subset, representative genes are presented on the heatmap. Cluster 5 is characterized by *GZMK*, *GZMA*, *CD27*, and *CXCR4*. **c** Representative markers for functionally unique *CD4*⁺ T cell subsets are presented in the dot plot. **d** GSEA revealing preferential enrichment of proinflammation and cytotoxicity (upper part of the heatmap) and DEGs (lower part of the heatmap) in effector-like (Clusters 2 and 4) and effector memory *CD4*⁺ T cells (Cluster 5) of cSjD. **e** The volcano plot presents DEGs in effector memory *CD4*⁺ T cells from cSjD compared to those from HC. Genes that are listed in d are highlighted in the box. **f** Gene ontology analysis using Cluster 5 cSjD DEGs reveals significant enrichment of gene set associated with cytolysis. Blue and red dot refers to gene ontology terms whose scaled fold enrichment is arbitrarily below and above 0, respectively. **g** The volcano plot shows statistically significant DEGs in *FOXP3*ʰⁱᵍʰ Tregs in cSjD compared to HC. Genes that are specific for cSjD but not identified in aSjD are highlighted with the box. **h** Dot plot reveals cSjD-related upregulation of thymocyte selection associated high mobility group box (*TOX*) in the *FOXP3*ʰⁱᵍʰ Tregs. Note the statistical significance of *TOX* between cSjD and non-cSjD. **i** While aSjD Tregs are

less immune-suppressive than HC Tregs, cSjD Tregs are as immune-suppressive as HC Tregs, analyzed by Treg suppression assay. Results were normalized to the proliferating T cells cultured without Tregs. A total of three patient samples for children (ages ranging from 12 to 18, newly diagnosed) and adults (ages ranging from 53 to 59, diagnosed for 1 or 5 years) group with a total of seven healthy controls (ages ranging from 15 to 54) to were used for the Treg suppression assay. Two independent experiments for each child and adult group were performed. Error bar refers to standard deviation. **j** Flow cytometric analysis of the mean intensity of phospho-STAT1 in IFN gamma-treated PBMC. Live CD45⁺CD3⁺CD4⁺ T cells are analyzed. Error bar refers to standard deviation. A total of three patient samples per group were subject to the experiment. One experiment for each child and adult group was performed. Dot or gene colored by blue and red refers to down and up-regulated DEGs, respectively, and a grey-colored dot refers to a gene that has no statistical significance in the differential expression (**e, g**). Statistical significance was obtained by comparing two groups of interest with a two-tailed unpaired Student's *t*-test. *p*-value < 0.01 ** (**i, j**). HC healthy control, cSjD childhood Sjögren's disease, BX biopsy-positive non-cSjD without RP, BXRP biopsy-positive non-cSjD with RP.

## cSjD contains a transcriptionally distinct subset of memory B and plasma cells

We identified 1,623 cSjD DEGs (Supplementary Data 11), 17 cSjD-related (Supplementary Data 12), and 118 cSjD-specific DEGs for B subsets (Supplementary Data 13). There was no remarkable difference noted in the cell cycle among the groups. B cell sub-clustering identified a total of ten

subsets, consisting of naïve B (Clusters 0, 1, 4, 5, 7 and 8), memory B (Clusters 2, 3, and 6), and plasma cell subsets (Cluster 9) (Fig. 3a). Among the subsets, Cluster 4 characteristically upregulated DNA nucleases *NEIL1* and *PLD4* (Fig. 3b). Cluster 6 was defined as memory B cells upregulated *CD1C*, *CLECL1*, *CD11C*, and *TNFRSF13B* compared to other clusters while Cluster 9 overexpressing *CD27* and *SLAMF7* was defined as plasma cells.

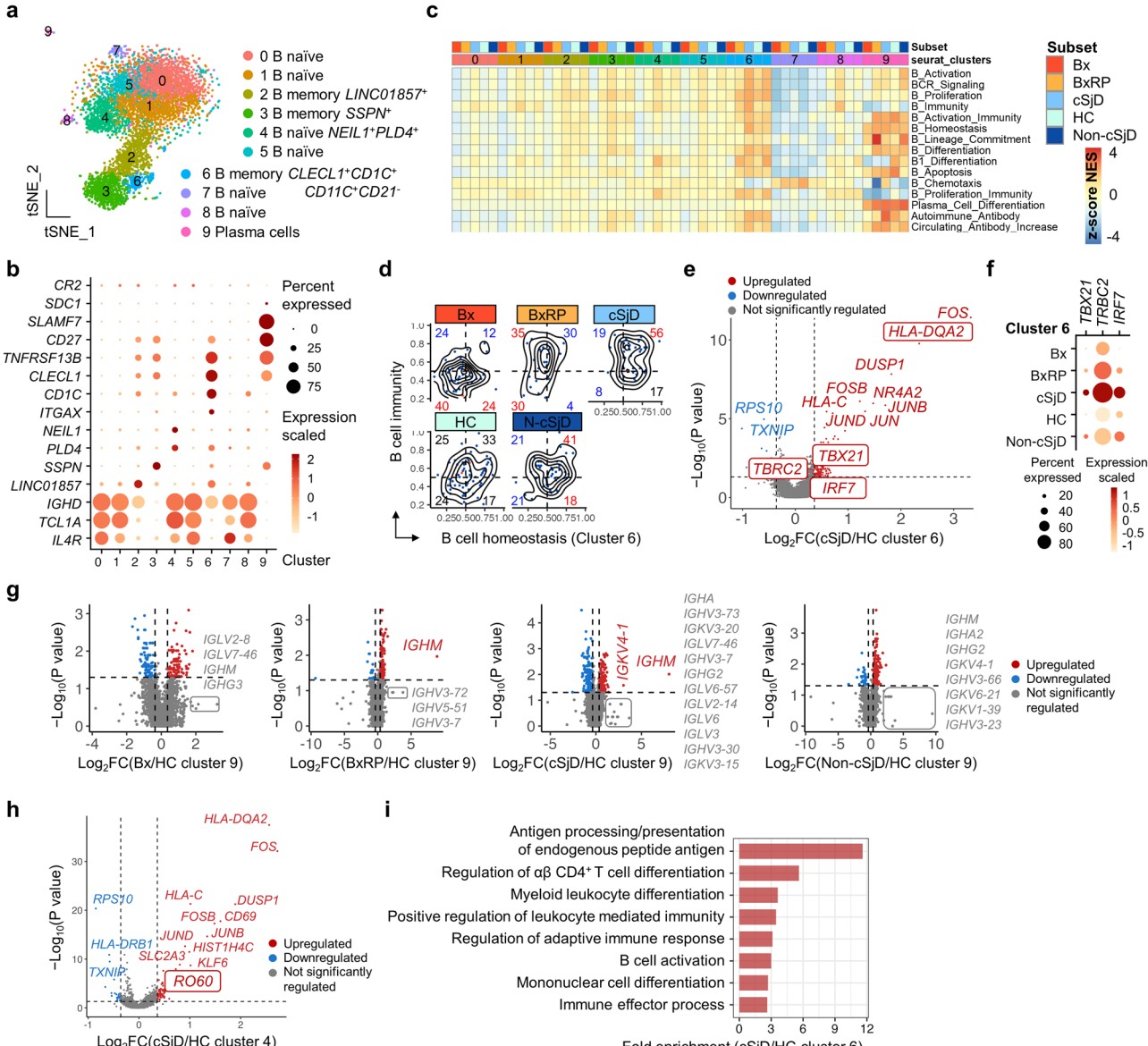

**Fig. 3 | Identification and characterization of B subpopulations by scRNA-seq.**
**a** A total of ten major B subsets with distinct canonical markers are presented on the tSNE plot. **b** Representative markers for functionally unique B subsets are presented in the dot plot. **c** GSEA reveals differentially enriched immune-associated signatures across the subsets and the groups. *CLEC1⁺CD1C⁺* B cells (Cluster 6) in cSjD preferentially upregulated gene sets associated with autoimmune antibodies. **d** Hex density enrichment plot reveals B cell homeostasis and immunity signatures in Cluster 6 across the groups. Note that the B cells in patients with cSjD and, to a lesser extent, non-cSjD show substantial skewness toward both types of signatures. The numbers in red and blue on each quadrant illustrate upward and downward trends, respectively, compared to HC. **e** The volcano plot reveals that the *CLEC1⁺CD1C⁺* B subset significantly upregulates *TBX21, IRF7, TRBC2,* and *HLA-DQA2* involved in B

cell immunity signature in cSjD compared to HC. Genes that are specific for cSjD but not identified in aSjD, are highlighted with the box. **f** Dot plot indicating the upregulation of *TBX21, IRF7,* and *TRBC2,* which is remarkably specific to cSjD (Cluster 6). **g** Immunoglobulin genes upregulated in plasma cells of cSjD and BxRP, compared to HC. **h** *PLD4⁺NEIL1⁺* naïve B cells (Cluster 4) shows upregulation of DEGs such as *Ro60.* **i** Gene ontology analysis reveals significant enrichment of gene sets associated with antigen presentation of endogenous peptide antigens in Cluster 6 of cSjD compared to HC. Dot or gene colored by blue and red refers to down and up-regulated DEGs, respectively, and a grey-colored dot refers to a gene that has no statistical significance in the differential expression (**e–h**). HC healthy control, cSjD childhood Sjögren's disease, BX biopsy-positive non-cSjD without RP, BXRP biopsy-positive non-cSjD with RP.

The cSjD-preferential enrichment of DEGs in distinct cellular processes, such as B cell homeostasis/immunity and autoantibody positivity, was mainly noted in memory B and plasma cells, respectively (Fig. 3c, d). Of note, memory B cells of cSjD significantly upregulated B cell immunity signature (*TBX21, IRF7, HLA-DQA2,* and *TRBC2*), which was absent in other groups, compared to HC (Fig. 3e, f). Plasma cells dramatically upregulated immunoglobulin genes (log₂-fold change > 1.5) with *IGHM* and *IGKV4-1* reaching statistical significance in cSjD and BxRP patients (Fig. 3g). Naïve *NEIL⁺PDL4⁺* B cells (Cluster 4) showed a cSjD-preferential enrichment in B1 B cell differentiation and proliferation-involved immunity

(Fig. 3c) along with Sjögren's syndrome antigen A2 (*Ro60*) upregulation (Fig. 3h).

Upon further analysis of memory B cells (Cluster 6), the enrichment of cSjD DEGs was noted in the modulation of immune cell activation and differentiation, leukocyte-mediated and adaptive immunity, vascular endothelial adhesion, and enrichment of gene sets such as IL-15 and IL-21 (Fig. 3i). In contrast, enrichment patterns in memory B cells between cSjD and non-cSjD were similar in B cell activation, BCR signaling, TLR, and Th1-associated cytokines, particularly IL-2, IL-18, IL-27, and TNF-α (Fig. 3c).

### *XCL1*⁺ NK cells are expanded significantly in cSjD compared to HC

We identified two NK (Clusters 1 and 7), a gamma delta T (γδ-T) (Cluster 2), and five *CD8*⁺ T (Clusters 0, 3, 4, 5, and 6) subsets (Supplementary Fig. 7a, b). Among NK subsets, Cluster 7 was consistent with activated and immunoregulatory phenotype, which is characterized by an XCL1^high NCAM1^high FCGR3A^low SELL^high ITGA1 ITGAE^low phenotype[35,36] (Supplementary Fig. 7c, d). More importantly, the fraction of *XCL1*⁺ NK cells expanded significantly in cSjD compared to HC, whereas cSjD patients showed a tendency of reduced % of effector *CD8*⁺ T cells (Cluster 5) compared to non-cSjD (*P* = 0.0556) (Supplementary Fig. 7e).

In addition, the ratio of *CD56*^high to *CD56*^hlow NK cells showed an increased tendency, especially in BxRP (Supplementary Fig. 7f). cSjD showed an increased tendency in chemokine expression, such as *CCL3* and *CCL4*, in NK cells, compared to Bx and BxRP (Supplementary Fig. 7g). Interestingly, a trend toward increased expression of *CCR1*, *CCR2*, and *CCR4* associated with disease progression was identified in various subsets of immune cells, especially in *CD14*⁺ monocytes and *CD4*⁺ T cells, and *XCR1*, the receptor of *XCL1*, was highly expressed in plasma B cells from cSjD (Supplementary Fig. 7h). Unexpectedly, circulating *CD8*⁺ T and γδ-T subsets showed only mild, sporadic enrichment patterns for several immune-related pathways (Supplementary Fig. 7i). However, when we predicted the biological functions using DEG-based gene ontology analysis in immune subsets from Biopsy⁺ compared to HC, few immune subsets, such as γδ-T and effector *CD8*⁺ T subsets, enriched gene sets associated with effector function, including γδ-T activation, which was driven by *CD247*, *JAML*, *TRDC*, *TRGC1*, or *KLRC1* upregulation (Supplementary Fig. 7j). Interestingly, the γδ-T cell enrichment was not identified in the same condition of non-cSjD, but in cSjD, compared to HC. A full list of cSjD, cSjD-related, cSjD-specific, and biopsy-related DEGs identified in each cluster compared to HC is available in Supplementary Data 14–17, respectively.

### cSjD exhibits distinct patterns of cell-to-cell interaction

As coordinated crosstalk among immune cell subsets is a hallmark of autoimmunity[37], cell-to-cell interaction was examined by applying the CellChat analysis (Fig. 4a). Among all groups, cSjD patients showed the strongest increase in both the number and strength of cell-to-cell interactions compared to HC (Fig. 4b). Many signaling pathways were conserved across the groups (Supplementary Fig. 8) except for CD48, PARs, and FASLG, which were uniquely present in cSjD. Strikingly, the interaction strength in almost all types of signal-receiving effector cells was found to be differentially increased in cSjD compared to HC (Fig. 4c). Of note, the interaction numbers of signal receiving naïve Tregs from B cells and myeloid cells significantly increased notably in cSjD (Fig. 4d).

In non-cSjD, only MAIT-like *CD8*⁺ cells showed increased interaction strength with non-myeloid/B cell types and activated Tregs (Supplementary Fig. 9a). In BxRP, IFN-related monocytes and activated Tregs showed increased interaction between them although overall minimal alteration was identified (Supplementary Fig. 9b). There was no remarkable interaction noted in Bx patients, except for pDC and resident memory-like *CD8*⁺ T cells (Supplementary Fig. 9c). SjD patients from the published dataset did show the enhanced interaction of *CD8*⁺ T subsets, including naïve and central memory subsets, compared to HC (Supplementary Fig. 10), which was absent in cSjD. Rather effector and MAIT-like *CD8*⁺ cells showed enhanced interaction in cSjD patients (Fig. 4c).

Effector memory *CD4*⁺ T cells in cSjD showed enhanced interaction with IFN-related monocytes, B cells, and DC, compared to other groups (Supplementary Fig. 11a). Compared to non-cSjD, both Bx and BxRP differentially increased γδ-T interaction, suggesting its potential importance in biopsy positivity (Supplementary Fig. 11b). Of note, when we combined Bx and BxRP into the biopsy positive (Biopsy⁺) group, plasma cells in Biopsy⁺ were identified to send and receive signals more frequently and strongly compared to non-cSjD (Supplementary Fig. 11b).

We also investigated ligand and receptor pairs (L–R) with a focus on the interactions with high communication probability. For example, *CLEC2C* (naïve B) and *KLRB1* (effector memory *CD4*⁺ T) were the L–R pair specific for cSjD compared to other groups (Fig. 4e). Ligand *CD4* on naïve Tregs was predicted to increase the interaction with many MHC II-related receptors on B and myeloid subsets in cSjD patients, compared to HC (Fig. 4f). *KLRB1* and *KLRK1* on the γδ-T from biopsy-positive groups were analyzed to interact with HLA-E and C-type (Ca²⁺-dependent) lectins, such as *CLEC2B* and *CLEC2D* expressed on the different immune cell types (Fig. 4g).

### Distinct DEGs in unique immune subsets noted in recurrent parotitis

To examine immune cell subsets that could potentially contribute to RP, one of the characteristic clinical features in cSjD[10–12,38], we regrouped and analyzed the sample datasets according to the presence of RP (Fig. 5a). First, altered frequencies of *BCL2*⁺ (Fig. 2a, Cluster 3) and *SOX4*⁺ (Fig. 2a, Cluster 6) naïve *CD4*⁺ T subsets were noted in cSjD patients with RP, compared to those without RP (Fig. 5b). The fraction of *SSPN*⁺ memory B cells (Fig. 3a, Cluster 3) tended to be suppressed along with effector *CD8*⁺ T cells in cSjD RP (Supplementary Fig. 6a, Cluster 5).

In Fig. 5c, *CLEC1L*⁺ memory B cells (left) clearly showed enrichment of gene sets in cSjD with RP patients compared to those without RP, including antigen presentation, IL-21, and IL-27. Similarly, DCs, *XCL1*⁺ NK cells, and effector memory *CD4*⁺ T cells showed RP-preferential enrichment with immune-related signatures. Interestingly, DCs in BxRP and cSjD RP patients concordantly increased enrichment with antigen presentation. Gene sets associated with activation of the immune response, such as proinflammatory, cytolytic granule, and poly(I:C), were more enriched in the *CD4*⁺ effector memory T cells in cSjD RP than in cSjD without RP (bottom, right). Moreover, patients with RP particularly had M1-like features in *CD14*⁺ monocytes, compared with those without RP (Fig. 5d).

A full list of RP-related DEGs identified in each cluster from BxRP and cSjD RP, compared to RP-negative groups, is available in Supplementary Data 18. Among DEGs, representative genes in RP showing significant differences compared to HC are presented in Fig. 5e. Interestingly, all these genes were derived from myeloid subpopulations. Moreover, upregulation of *CCL3* and *TRDC* was associated with RP (Fig. 5f). Furthermore, the *CCL3* expression in conventional NK cells was specific for cSjD RP, compared to non-cSjD. We also investigated the CCL3 receptor CCR expression and identified upregulation of *CCR5* in MAIT-like *CD8*⁺ T cells in both BxRP and cSjD RP patients, compared to the corresponding non-RP groups (Fig. 5g).

Next, we analyzed cell-to-cell interactions to identify an RP-relevant interaction pattern, which was more remarkable in BxRP than cSjD (Supplementary Fig. 12). DCs and *CD8*⁺ effector T cells were top-ranked to have strong interaction with each other in BxRP patients (Supplementary Fig. 12, top, right), compared to Bx, which was also partially notable in cSjD RP patients (Supplementary Fig. 12, bottom, right). In cSjD with RP, B and myeloid cell subsets—in particular the pDC and plasma cells—were differentially sending signals to various immune subsets, including *CD8*⁺ effector T and activated NK cells (Supplementary Fig. 12, bottom). Among these interactions, MHC class I molecules upregulated in many immune subsets were predicted to be significant ligands for *CD8*⁺ effector T and activated NK cells in BxRP, which includes *HLA-C:CD8A* and cSjD RP (Fig. 5h).

### Discussion

cSjD is an autoimmune disorder, necessitating a better understanding of the etiology at molecular and biological levels. By applying an unbiased and comprehensive analysis of PBMC by scRNA-seq, we identified the key immune subsets that may play critical roles in cSjD pathogenesis. These subsets include IFN-related *CD14*⁺ monocytes, *XCL1*⁺ NK cells, effector memory *CD4*⁺ T cells, and *CLECL1*⁺ memory B cells with upregulated

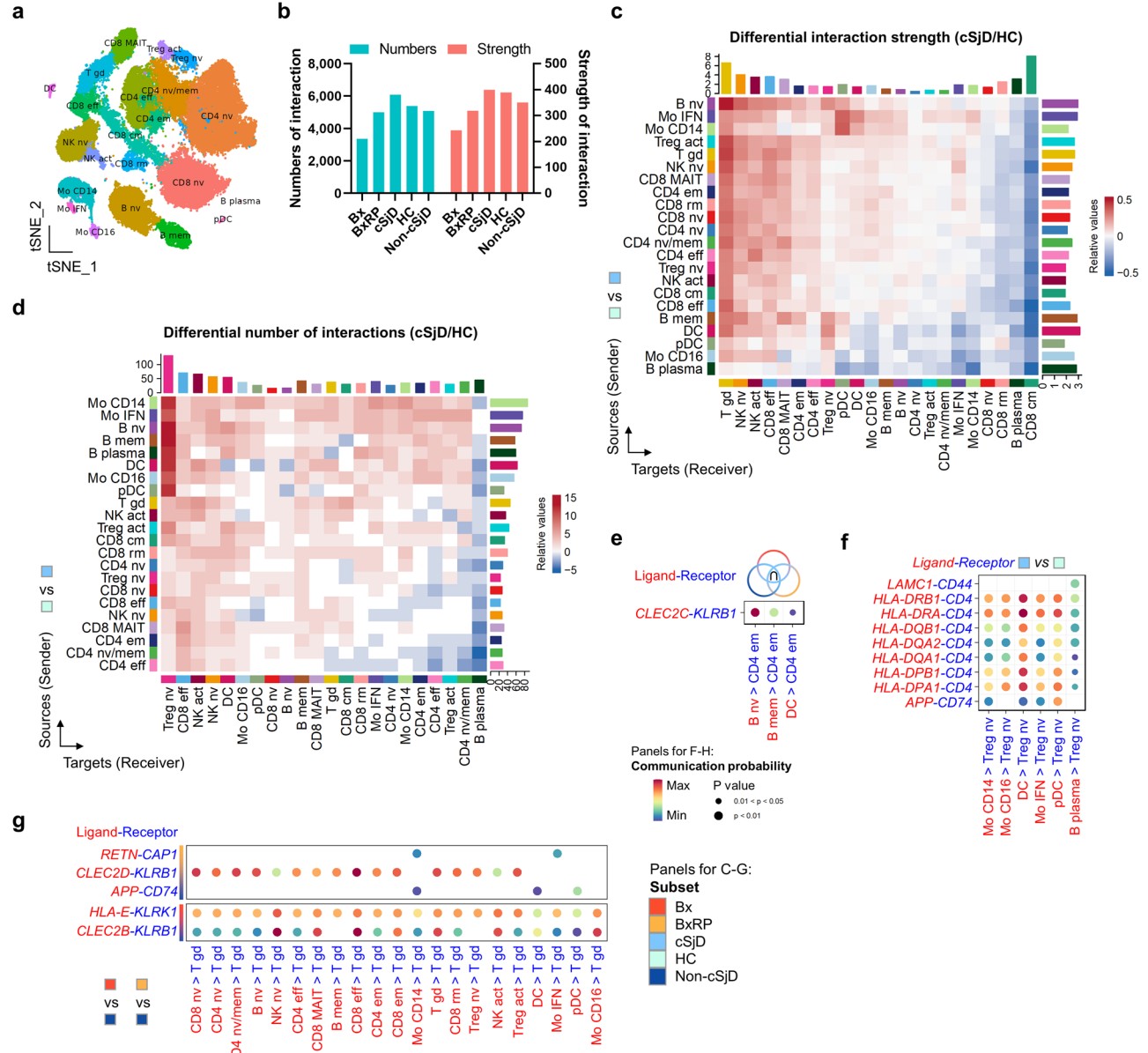

**Fig. 4 | Comprehensive analysis of scRNA-seq data by CellChat for cell-to-cell interactions. a** Immune cell subsets analyzed by CellChat in the original tSNE plot. **b** Enhanced number of interactions and interaction strength noted in cSjD. Differential interaction strength **c** and numbers **d** are presented on the heatmap. The overall increase in the interaction strength in the signal-receiving effector immune subsets, along with a dramatic increase in the interaction numbers noted between *FOXP3*[low] Tregs (or naïve Tregs) and myeloid/B subsets, in cSjD compared to HC. Bars refer to the sum of the computed interaction strength or numbers in each column and row. Relative values = the interaction strength/number from source to target in cSjD−the interaction strength/number from source to target in HC. **e**–**g** Identification of significant ligand and receptor pairs between immune subsets of interest. **e** The communication probability of CLEC2C ligand and KLRB1 receptor genes are significantly upregulated between B and effector memory *CD4*[+] T cell subsets in cSjD. **f** *FOXP3*[low] Tregs show increased communication probability with myeloid subsets through CD4 and MHC class II-related genes in cSjD, compared to HC. **g** Pairs of CLEC ligands and KLRB receptor genes show increased communication probability in γδ-T cells of Bx and BxRP, compared to non-cSjD. Mo monocytes, NK natural killer, eff effector, em effector memory, mem memory, cm central memory T cell, nv naïve, pDC plasmacytoid dendritic cells, MAIT mucosal-associated invariant T, act activated or *FOXP3*[high], IFN interferon-related, HC healthy control, cSjD childhood Sjögren's disease, BX biopsy-positive non-cSjD without RP, BXRP biopsy-positive non-cSjD with RP.

*TBX21*. More importantly, we also proposed the mechanisms by which RP is initiated and progresses.

Effector CD4[+] T cells play an important role in host defense against microbial infection, anti-tumor immune response, and pathology of autoimmune diseases[39], including aSjD. Similarly, cSjD effector memory *CD4*[+] T cells upregulated key transcriptional features of effector cytotoxic functions such as *GZMA* and *SELL*, known to be important in aSjD and rheumatoid arthritis[32,40]. Notably, this trend was highly specific for cSjD among all groups analyzed, as evidenced by our DEGs, functional enrichment, and interactome, implying their potential roles in mediating cytotoxic damage in

cSjD. Thus, future investigations of T cell receptor repertoires of effector CD4[+] T cell subsets in the periphery and the target glandular tissue will provide insights into pathogenic T subsets in cSjD.

Loss of Treg functionality has been documented in autoimmune conditions, including aSjD[34,41]. What appears to be unique in cSjD is that Tregs from cSjD are functionally intact and that *TOX* expression is upregulated in cSjD Tregs as opposed to its expression in aSjD Tregs when compared to HC Tregs. To our best knowledge, only one aSjD Treg functional assay study has been published[34], which indicates the impaired suppressor activity of these cells. Our current analysis further supports the

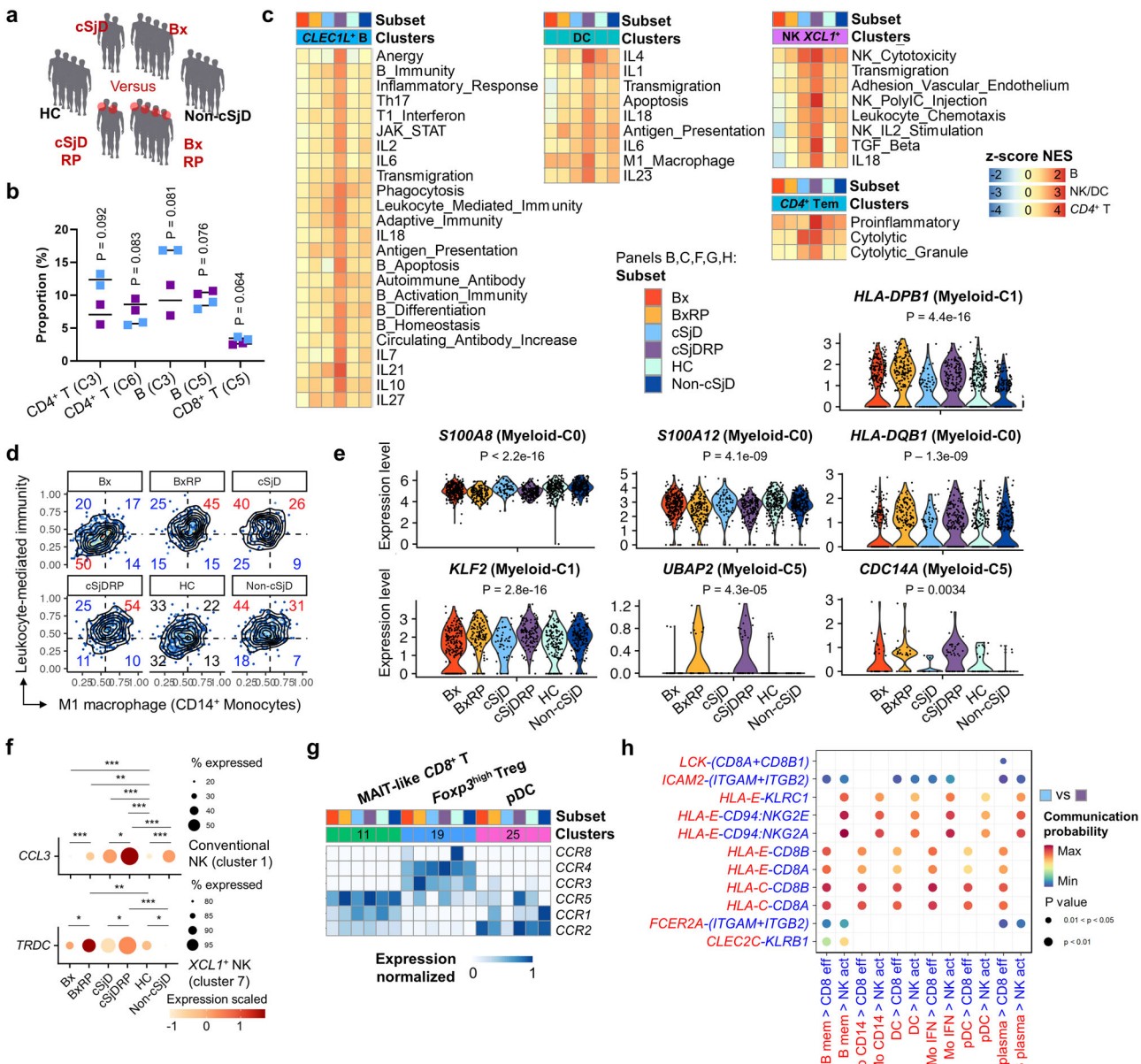

**Fig. 5 | Identification and characterization of immune subsets involved in recurrent parotitis. a** The pediatric cohort is sub-classified by the presence of RP, which is indicated by the red circle. BioRender software was used to create the scientific images under an academic license. **b** Proportion changes of selected immune subsets in two RP positive and two RP negative cSjD individuals. **c** GSEA shows preferential enrichment patterns for RP in different subsets. **d** Hex density enrichment plot revealing enrichment pattern of M1-like macrophage and leukocyte-mediated immunity in the classic monocyte subsets (Cluster 0, 1, and 4) across the groups. RP exhibits a more inflammatory phenotype than those without, as evidenced by the RP-preferential shift toward the inflammatory signatures. The numbers in red and blue on each quadrant illustrate upward and downward trends, respectively, compared to HC. **e** Representative genes showing significant upregulation in BxRP and cSjD RP patients, compared to those without RP, are found to be

present in the myeloid clusters. These genes were not significantly dysregulated in non-cSjD when compared with HC. **f** Dot plot presenting RP-related upregulation of *CCL3* and *TRDC* genes in NK subsets. Statistical significance was obtained by comparing two groups of interest with a non-parametric Wilcoxon rank-sum *t*-test. \**p* < 0.05, \*\**p* < 0.01, and \*\*\**p* < 0.001. **g** Expression pattern of CC chemokine receptors. Clusters present in the heatmap refer to immune cell subsets in the original tSNE plot as shown in Fig. 1e. **h** Identification of significant ligand and receptor pairs between immune subsets of interest, including effector *CD8*[+] T and *XCL1*[+] NK subsets. These subsets have shown increased communication probability with B and myeloid subsets via MHC class I-related genes in cSjD RP, compared to the groups without RP. HC healthy control, cSjD childhood Sjögren's disease, BX biopsy-positive non-cSjD without RP, BXRP biopsy-positive non-cSjD with RP, cSjDRP cSjD with RP.

results from aSjD while revealing that cSjD Tregs maintained their function in an in vitro analysis. Presumably, TOX, a DNA-binding protein essential for CD4 cell lineage development[42] and one of the well-known markers in T cell exhaustion[33,42], may explain this finding. TOX can also be upregulated by inflammatory cytokines in a manner that is uncoupled from T cell exhaustion[43]. In addition, molecules such as *JUNB* and *CTLA4* were also upregulated in cSjD Tregs but not in aSjD Tregs. Whether their upregulation in cSjD Tregs also dictates the intact suppressive activity of Treg in

addition to *TOX* compared to aSjD Tregs is one of the key questions in our future study.

To our best knowledge, the *CD14*[+]*CD16*[−] monocyte subset with the prominent expression of both type I and type II is a unique discovery noted in cSjD. This particular subset of monocytes may explain underlying etiologic factors in cSjD, such as viral infection[44], enhanced transmigration into target glandular tissues[45], immune cell recruitment[46], and maintenance and replenishment of antigen-presenting cells[47]. Although viral infections can

trigger immune dysregulation in autoimmune disorders, DEGs involving viral infection appear to be absent in cSjD compared to non-cSjD. A varying degree of genetic and epigenetic susceptibility in an individual at the time of viral insults may determine the risk for cSjD onset and progression rather than the absolute presence and absence of infection itself. Additional plausible mechanisms by which IFN-related monocytes shape cSjD may lie in their impact on leukocyte recruitment to target tissues, based on the upward trend of *CCL2* expression on these cells in Bx, BxRP, and cSjD (all sharing biopsy positivity), but not in non-cSjD, compared to HC. Type I IFN-driven CCL2 is known to be important for recruiting various immune cell types, including monocytes, NK, and T cells, to tissues[48,49]. Thus, IFN-driven chemotaxis in monocytes involving CCL2 may be a key biological process in cSjD, distinct from non-cSjD.

Transcriptional features of *XCL1*+ NK cells, which expanded in cSjD, support immune modulatory phenotype[35,36]. A study reported an anti-inflammatory role of TRAIL+ NK cells through death receptors in a chronic murine cytomegalovirus infection model for SjD-like condition[36]. However, our analysis indicates that *XCL1*+ NK cells interact with various immune cell subsets independent of the death receptors. As XCL1 is known to be involved in the secretion of IFN-γ, IL-22, and TNF-α, which is important for crosstalk between NK cells and salivary epithelial cells[50–52], XCL1+ NK cells in cSjD may play more pro-inflammatory rather than anti-inflammatory roles following the recruitment to the inflamed exocrine glands, based on the preferential enrichment of transmigratory gene sets. Consistent with the study reporting pDC and NK cell localization in close proximity to the inflamed tissues[50], the *XCL1*+ NK: pDC interaction was identified in cSjD patients with RP in our cell-to-cell interaction analysis, and *CCR5* expression was increased in pDC from cSjD. In addition, *XCR1*, the cognate receptor for *XCL1*, was specifically upregulated on plasma cells from cSjD, suggesting a potential cellular interaction between these two subsets.

The *TBX21(T-bet)*+ B cells, which are characterized not by CD21 but by CD11c expression, are identified in mice and humans in a variety of disease contexts, including autoimmunity[53]. T-bet+ B cells were present in aSjD[54,55] along with autoreactive CD21−/lowCD86+CD69+ memory B cells[55]. *CLECL1*+ memory B subsets with *T-bet* overexpression in cSjD upregulated *CD69, CD83*, and *HLA-DQA2*, which signifies activated phenotype. Despite early defective tolerance checkpoint and autoantibody production in peripheral B cells reported in an aSjD patient[56], the majority of randomly selected cSjD from our cohort happened to be anti-Ro/SSA autoantibody negative, which is also reflected in our scRNA-seq dataset. T-bet is known to promote antibody-secreting cell differentiation by limiting the inflammatory effects of IFN-gamma on B cells[57]. Because high enrichment of IFN signaling was noted in the *CLECL1*+ memory B subset in cSjD patients, the extent by which *TBX21*+ influences plasma cell differentiation and autoantibody production under the pro-inflammatory environment in cSjD is still unclear. In addition, although the number of anti-SSA-positive patients is relatively low in cSjD compared to aSjD[10], future studies on the anti-SSA-positive patients compared to the autoantibody-negative patients in cSjD will answer the role of the B cell subset in this condition.

Expanded circulating activated CD8+ T subsets in aSjD[58] and reduced frequency of highly cytotoxic effector CD57+CD27−CD45RA+CD8+ T cells along with circulating DCs in the peripheral blood of aSjD[59] have been reported. Interestingly, a reduction in the frequency of the effector *CD8*+ T subset was identified in cSjD compared to non-cSjD. It should be noted that contrary to non-cSjD, the effector *CD8*+ T subset in cSjD was enriched with transmigration but not with vascular endothelial adhesion, suggesting potentially inefficient homing of the effector T cells to the tissue. In addition, cytotoxic CD8+ T cells not only damage the target tissue in aSjD patients[60] but also play an integral role in viral clearance in the peripheral blood of aSjD[59]. Furthermore, MAIT-like CD8+ T cells in the blood are suggested to transmigrate into the affected salivary tissue[61,62]. Contrary to a previous study on aSjD[62], MAIT-like CD8+ T cells in cSjD patients were not naïve but transcriptionally activated, as evidenced by enrichment with IL-23 in our study. Thus, MAIT-like CD8+ T cells are presumed to play a role in cSjD pathogenesis through the acquisition of the Th17-like phenotype in

response to IL-7 and IL-23[61]. Taken together, profiling CD8+ T cells in the periphery as well as in the foci of biopsy tissues from cSjD patients compared to non-cSjD or aSjD will be necessary to examine their impact on organ damage.

To understand salivary gland inflammation presented as RP, we provide molecular evidence that RP may be attributed to effector immune cells, such as effector *CD8*+ T cells, effector memory *CD4*+ T cells, and *XCL1*+ NK cells, potentially interacting with *CLEC1L*+ memory B cells, *CD14*+ monocytes, and pDC. The spatial proximity of pDC and activated NKp46+ NK cells in aSjD patients mentioned earlier[50] supports our findings. Given the seronegative clinical features (normal IgG level and negative anti-Ro/SSA) in RP patients, we speculate that the *CLEC1L*, which encodes a member of the C-type lectin-like domain superfamily, expressed on memory B cells, may contribute to recognizing endogenous and exogenous ligands for cellular and inflammatory responses by serving as pattern recognition receptors. The upregulation of *TRDC* (T cell receptor delta constant), which participates in antigen recognition, in the activated NK subset (*XCL1*+ NK) in our scRNA-seq database and other datasets[63,64], may be associated with dysregulated innate immunity in RP as high innate-like lymphoid cells[65]. Preferential enrichment of M1-like polarized *CD14*+ monocytes in RP patients, compared to the non-RP or non-cSjD group, also supports our notion. CCL3-upregulating NK cells might also be a causative factor for RP, recruiting immune cells, such as CCR5-expressing MAIT-like CD8+ T cells, which are transmigratory[61,62]. A trend toward decreasing frequency of the effector *CD8*+ T cells associated with RP in cSjD is noted. A body of emerging evidence has supported the role of preferential migration of pathogenic CD8+ T subsets into glandular tissue in aSjD[59,66].

Identification of RP-related genes in myeloid subpopulations highlights their clinical relevance. *UBAP2* is responsible for the ubiquitylation and degradation of RNA polymerase II, which prevents the transcription of damaged RNA following UV exposure[67]. Likewise, *CDC14A* is upregulated following DNA damage in patients with periodontitis[68]. Interestingly, *UBAP2* is highly expressed in a subset of bone marrow-derived monocytes with the potential for osteoclast lineage differentiation[69]. Based on the upregulation of these genes associated with nucleic acid damage in myeloid cells in RP+ subjects, we propose that myeloid cells, such as monocytes and pDC, may harbor abnormalities in nucleic acids within the nucleus or mitochondria, which could be an inflammatory or differentiating cue by mimicking microbial or viral infection in patients with RP. Potential interaction between pDC and activated NKp46+ NK cells in aSjD patients has been reported[50]. RP-related monocyte activation markers, such as *S100A8, S100A12*, and *HLA* gene variants, might also support proinflammatory processes and increase the risk of developing cSjD, as suggested in aSjD[70–73]. Enhanced antigen presentation capacity driven by monocyte-to-macrophage differentiation and/or defective phagocytosis could be involved in patients with RP, which warrants further studies.

Our study highlights the importance of cell-to-cell interactions in cSjD. Whether IFN-related monocytes directly activate effector memory CD4+ T cells in cSjD is unclear. However, several studies have supported its possibility. IFN-α can induce and maintain costimulatory molecules, such as CD80, CD83, CD86, and HLA-DR, on CD14+ monocytes[74], similar to what is found in our study. The IFN-α-treated CD83+CD14+ non-dendritic monocytes play a role in antigen presentation and autologous memory CD4+ T cell proliferation in inflammatory bowel disease[74]. Similarly, MHC-II+ monocytes interact with effector CD4+ T cells via antigen presentation in an intravascular niche, leading to autoimmune glomerulonephritis in a mouse model[75]. IFN-related monocytes and effector *CD4*+ T cells in cSjD were found to share proinflammatory cytokine signaling, such as TNF-α, IL-15, and IL-27, which is reported to be critical in the local inflammatory niche in aSjD[76–78]. However, it should be noted that our interactome results can be limited by the intrinsic nature of the cell-cell interaction methodology, e.g., inference, which is based on the experimentally validated ligand–receptor database.

One of the limitations in our study stems from the absence of the cSjD-specific diagnostic criteria. Our clinical observations and laboratory findings support the possibility that cSjD and aSjD are likely to be distinct disease

entities[6,38]. The establishment of the cSjD-specific criteria, reflecting the importance of the most common clinical presentation of RP, is urgently needed in the field. Another limitation is the absence of information on biopsy tissues in the study since the complete understanding of cSjD pathogenesis requires the profiles and the functions of immune cell subsets in the target tissues. For example, we only found minimal transcriptional changes in Biopsy[+], suggesting potential involvement of innate immune sensing of autoantigens by IFN-related monocytes and effector $CD8^+$ T cells in this entity. In addition, we did not see much evidence that DC and pDC are active contributors to cSjD pathogenesis, at least in the periphery. The fraction of circulating pDC is known to be decreased[5,79] while their roles in the target tissues as the main driver for the type I IFNs are well accepted in aSjD. Therefore, investigation of tissue-infiltrating pDC is needed to properly conclude their roles in cSjD pathogenesis. Finally, we presume that biopsy-negative non-cSjD (symptomatic) may have an unidentified auto-immune condition, which is to be further elucidated. Nevertheless, we postulate that *PLCG2* could be uniquely involved in biopsy-negative non-cSjD, e.g., by regulating monocyte and macrophage plasticity, because PLCG2 is known to be important in determining myeloid lineage commitment and different functionality of monocytes[80,81].

Altogether, cSjD demonstrates distinct proinflammatory immune cell subsets, characterized by frequent and enhanced interaction amongst effector immune cell subtypes. Non-cSjD displays monocyte heterogeneity pertaining to inflammatory properties with relatively increased immune cell interplay. Biopsy-positive non-cSjD presents proinflammatory M1-like polarized monocytes, while immune cell interaction is barely present. Patients with RP have more inflammatory immune subsets, including $XCL1^+$ NK cells and their enhanced interaction. Therefore, we propose that enhanced immune cell interaction and monocyte heterogeneity may serve as a differentiating factor for cSjD from non-cSjD. A subset of more activated NK cells was noted in patients with RP. Tregs from cSjD are functionally competent, which necessitates prospective, longitudinal studies to gain insights into how activated immune cells bypass suppression by Tregs in cSjD patients.

## Data availability
The raw and processed data from scRNA-seq in this study is available in the NIH dbGAP database with controlled access under the accession number phs003048.v1.p1. Adult SjD scRNA-seq data is available in a publication by Hong et al.[14]. PBMC scRNA-seq data from healthy children and adults is available in GEO Datasets with accession numbers of GSE148633, GSE206295, GSE168732, GSE148633, GSE216489, GSE211560, and phs003048.v1.p1. PBMC scRNA-seq data from centenarians is available in the NBDC Human Database with accession number NBDC hum0229.v1. The source data behind the graphs in the figures can be found in Supplementary Data 20.

## Code availability
Details of publicly available software used in the study are given in the "Methods" section. Codes used in our analysis are available at GitHub repository https://github.com/mckimcd177/Integration and https://doi.org/10.5281/zenodo.10654448[82].

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

## Acknowledgements

The current study was supported by the High Impact Research Grant from the Sjögren's Foundation, NIH/NIAMS AR079693, and NIH/NIDCR DE032707 (all to S.C.) as well as NIH/NIDCR DE023838 (S.C. and Q.Y.). W.Z. was supported by NIH grants (CA203834, CA260239, AR079693) and an endowed fund from the Dr. and Mrs. James Robert Spenser Family at the University of Florida. We thank the University of Florida ICBR NextGen DNA Sequencing Core Facility for assisting in designing and performing library preparation and NextGen DNA sequencing, which is partially subsidized by the University of Florida Health Cancer Center. We also thank Nicole Winn for providing de-identified information on the subjects and Jimin Yoon for the generation of the graphic summary of our study groups using Adobe Illustrator and the related Supplementary Table 1.

## Author contributions

W.Z. and S.C. conceptualized, designed, and supervised the study. M.-C.K., S.C., and W.Z. developed the methodology. M.-C.K., U.D., J.P., I.B., L.W., and A.T. acquired specimens and data. M.-C.K. and N.B. performed bioinformatic analysis. M.-C.K., T.D., A.T., S.C., and W.Z. analyzed and interpreted the data. M.-C.K., Q.Y., R.K., A.T., W.Z., and S.C. wrote, reviewed, and revised the manuscript.

## Competing interests

The authors declare no competing interests.
