## [Peer Review File · Communications Biology]

Reviewers' comments:

Reviewer #1 (Remarks to the Author):

Sjögren's disease (SjD) is a chronic autoimmune disorder affecting the exocrine glands characterized by the presence of lymphocytic infiltrates of the affected glands and accompanied by the loss of organ's secretory function resulting in its typical symptoms like dry eyes and mouth (sicca). Although the disease typically affects adult women it can also manifest in childhood at which stage it remains difficult to diagnose due to a distinct than in the adult range of symptoms, like parotitis. Although this implies the existence of mechanistic differences between childhood and adult SjD, this subject remains obscure. Therefore, studies addressing this topic, like that by Kim et al. are of high interest not only to rheumatologists but also to a broader audience of immunologists. In their study, Kim et al. performed a thorough in-silico analysis of single-cell RNA sequencing datasets of peripheral blood leukocytes from patients affected by childhood SjD (cSjD) describing several transcriptional and inferred putative functional and interaction differences between the immune subsets of diseased and healthy conditions. The authors make a claim that they have identified differences between childhood and adult forms of the disease. The first difference being the identification of a distinct subset of monocytes characterized by high interferon signaling that the authors conclude is specifically present in cSjD but not aSjD. Another difference between the conditions put forward in the manuscript is normal in vitro Treg function in cSjD compared to decreased function of Tregs in aSjD. In addition, the authors report several transcriptional differences in immune cell subsets that are present in cSjD compared to healthy controls, like an upregulated inflammatory signature of effector CD4 T cells, and dominance of NK cells with upregulated activation phenotype. Despite the importance of the topic, the major claims of the study are weakened by issues concerning experimental design, interpretation and lack of validation.

Major points:

1. The claim about the identification of a distinct subset of monocytes characterized by high interferon signaling that is specifically present in cSjD but not aSjD is based on parallel analysis of a publicly available scRNAseq dataset on blood leukocytes from aSjD patients published by a different group. However, the interferon-high subset appears to be present in both healthy and diseased samples in the cSjD dataset while being completely absent from the other dataset. If the interferon-high subset is present in healthy controls from cSjD dataset but is absent in healthy controls from aSjD dataset it implies that the difference reported is a difference between the datasets (perhaps relating to age or ethnicity of the subjects involved) but not between cSjD and aSjD. Are there any differences that are specific to cSjD that go beyond the features already reported for aSjD like increased IFN-signatures in immune cells including monocytes, upregulated inflammatory signatures on various leukocyte subsets or expansion of effector CD4 T cells that were reported in references 14-17?
2. Regarding the conclusion about differences in Treg suppressive activity in the conditions of aSjD and cSjD, is a decrease in Treg function in vitro an established trait of aSjD or is this shown for the first time in this study? In any case, the authors should provide information on how many aSjD patients were tested, their age, number of years from diagnosis as well as the sample size and how many times this experiment was repeated (I could not find this information in the manuscript). The authors state on page 9 that "many autoimmune diseases are associated with gradual loss of Treg function". Could it then be

that in the first years after diagnosis Treg functionality is normal in both aSjD and cSjD and only later declines which would be seen in older aSjD patients but not in young cSjD patients? Can the authors ascertain that this is not the case?

3.The conditions studied/groups compared are confusing. The study compared five groups: HC (healthy control), cSjD, Non-cSjD, Bx, and BxRP. As far as I could understand the authors classified as cSjD patients that fulfilled all the diagnostic criteria for aSjD. However, when considering cSjD the expert diagnosis may fall even when the 2016 ACR/EULAR criteria aren't fulfilled (like stated in Ref. 7) and the most common non-sicca symptoms are arthralgia and parotitis. Were the Bx and BxRP conditions diagnosed as cSjD by a physician and if not, as what were they diagnosed since they show an overlapping set of symptoms with the patients classified as cSjD (Suppl. Fig. 1). What is the meaning of having all the different groups if the main conclusions are based on identified differences between cSjD and HC? Throughout the manuscript the authors occasionally report of differences that are selectively present in one of the other groups but without explaining what it may mean. Furthermore, how is the "non-cSjD" group different from healthy control?

4.Could it be more informative for a study that attempts to identify differences in cSjD versus aSjD to continually keep the cSjD patients with and without parotitis as separate groups and include in the study an internally generated aSjD dataset for direct comparison?

5.The authors are encouraged to improve the strength of their claims overall through the study by performing validation experiments for the expression of proteins that they consider to be functionally implicated in the mechanism of the disease.

Reviewer #2 (Remarks to the Author):

Kim et al. describe several well-characterized cohorts of childhood Sjögren Syndrome and individuals which do not meet criteria for cSjD, and compare data to that of publicly available adult SjD. Notably, the authors detect a distinct pro-inflammatory monocyte cluster and intact regulatory T cell populations, both distinguishing cSjD from non-cSjD and adult SjD. In addition, sc RNAseq data implicates XCL1+ NK cells, effector memory T cells and a specialized memory B cell cluster in cSjD.

Overall, the presented manuscript offers technically sound analyses of immune subsets in a rare patient group. This is an important, well-performed study with a well-defined study cohorts and extensive bioinformatic work-up of a large single cell RNAseq data set. Some of the data is confirmed in flow cytometry and functional Treg assays. The presented data is a valuable resource for the community. However, the presentation of results may need some focus, clarification and re-structuring.

Specific comments:

1. Definition of clinical cohorts is quite complex. I suggest adding a graphical representation of cohort definitions for easier readability. If possible, simpler names for the three non-cSjD cohorts might be introduced.
2. Line 116-122: Description of results from Suppl. Fig. 4 at this point in the MS seems odd and stops

reading flow. M1/M2 polarization terminology has largely been abolished.

3. Figure 1: This figure is quite busy and could be streamlined. To illustrate description of results in the manuscript text, cluster presence in each clinical cohort should be shown (similar to Supplementary Fig 6E). Smaller data figures (Fig 1 H, I, J) could be moved to supplements instead. Fig 1L seems redundant, as data is presented in Fig 1M.

4. Suppl. Fig. 2J offers interesting insights into relative cellular abundance in the different cohorts and could be presented as a main figure. In addition, I suggest differential neighborhood abundance testing to potentially identify previously unknown cellular players in different disease cohorts or in HC vs. cSjD(+/-non-cSjD) in a less supervised fashion.

5. Suppl. Fig. 4: Datasets from adult SjD lack specific IFN-related monocytes - however, there is no direct DEG comparison with data from cSjD. This limitation should be discussed in the manuscript.

6. Suppl. Fig. 6.: Results section to this supplement is quite long, text could be limited to most important findings.

7. Figure 4: As the analysis method is only able to predict putative cell-cell interactions, the theoretical nature of presented cellular interaction data should be stated in the manuscript.

8. Figure 4: Text of figure Panels C, D and E is too small and therefore illegible. A different way of presentation should be chosen.

9. Figure 4E: Bx and BxRP were grouped into one biopsy positive group. Why was this grouping not performed for other analyses shown in Fig 1-3?

10. Line 825: no red circle is visible in Fig 5A.

11: Fig 5G: see comment 8

12: Figure 6 could be included in Fig 5.

13. Line 369-371: Statement not suitable for data manuscript, should be rephrased.

14. Can the authors further comment on the criteria that were not met in non-cSjD and do the authors conclude that patients in the non-cSjD cohort present with a distinct disease or a combination of different diseases?

15. Can the authors extrapolate from their data a specific molecule or migration receptor of blood leukocytes that is associated with RP and could be used diagnostically?

16. The presented data is a valuable resource for the community and therefore should be made available via a public data database (if personal rights allow).

AUTHORS' RESPONSE TO THE REVIEWERS' COMMENTS

We sincerely appreciate the reviewers' invaluable comments and suggestions (in blue), which certainly strengthened our manuscript (MS) during our revision process. The summary of the comments and our responses (in black) are listed herein. **“Response” Figures refer to the Figures presented in this document.**

Reviewer#1

Q#1. The interferon-high subset appears to be present in both healthy and diseased samples in the cSjD dataset while being completely absent from the other dataset. If the interferon-high subset is present in healthy controls from cSjD dataset but is absent in healthy controls from aSjD dataset it implies that the difference reported is a difference between the datasets (perhaps relating to age or ethnicity of the subjects involved) but not between cSjD and aSjD. Are there any differences that are specific to cSjD that go beyond the features already reported for aSjD like increased IFN-signatures in immune cells including monocytes, upregulated inflammatory signatures on various leukocyte subsets or expansion of effector CD4 T cells that were reported in references 14-17?

R#1. We investigated if a subset of type I and type II IFN-related monocytes is specific to cSjD or if the profile is related to the age or ethnicity of the subjects by analyzing published scRNA-seq data sets on healthy participants.

Various age groups of healthy PBMC datasets, including 4 groups of 11 children (GSE148633, GSE206295, GSE168732, and our dataset), 3 groups of 9 adults (GSE148633, GSE216489, and GSE211560), and a group of 2 centenarians (NBDC hum0229.v1), were subject to integrated analysis and sub-clustering of CD68⁺ myeloid cells (**Response Figure 1A**). We found that cluster 7 is IFN-related monocytes (**Response Figure 1B, C, and D**). Top 10 DEGs defining cluster 7 were MX1, IFI44L, MX2, ISG15, IFIT3, XAF1, PARP14, IFIT2, RNF213, and APOBEC3A (**Supplementary Figure 5E and Response Figure 1D**).

In addition, this cluster was also identified in PBMC samples with different ages and ethnicity groups of monocytes (Canada, China, and USA), supporting that IFN-related monocyte clusters are present regardless of age (**Response Figure 1C**) and ethnicity (**Response Figure 1E**).

However, what's fascinating to us is that GSEA shows that these IFN-related monocytes in healthy children clearly lack proinflammatory and M1 markers (**Response Figure 1F, Cluster 7**). Even in the same ethnicity of paired healthy children and adult monocytes (scRNA-seq dataset, GSE216489, **Response Figure 1F, blue box**). In other words, enriched proinflammatory signatures in the IFN-related monocytes are lacking in both healthy children and adults. IFN-related monocytes from adults rather showed slightly increased enrichment in anti-inflammatory signatures compared to those from healthy children. In summary, there was no remarkable difference in the enrichment pattern of proinflammatory gene signatures in IFN-related monocytes from healthy children and healthy adults.

In contrast, when we included 4 cSjD PBMC samples from our study into the integrated scRNA-seq datasets, enrichment scores of types I and II IFN gene signatures in IFN-related monocytes were found to be significantly increased in cSjD compared to HC children (**Response Figure 1G**).

Thus, based on these findings, we conclude that IFN-related monocytes may exist in both healthy child and adult individuals, but these monocytes are not enriched with proinflammatory gene signatures, whereas the cSjD IFN-related monocyte (cluster 4) is highly proinflammatory in gene signature. We included this important discovery in **Supplementary Figures 5D-5I** and described in the main body of the MS (**Yellow-highlighted**) (**Page 8**).

Q#2. Regarding the conclusion about differences in Treg suppressive activity in the conditions of aSjD and cSjD, is a decrease in Treg function in vitro an established trait of aSjD or is this shown for the first time in this study? In any case, the authors should provide information on how many aSjD patients were tested, their age, number of years from diagnosis as well as the sample size and how many times this experiment was repeated (I could not find this information in the manuscript). The authors state on page 9 that “many autoimmune diseases are associated with gradual loss of Treg function”. Could it then be that in the first years after diagnosis Treg functionality is normal

in both aSjD and cSjD and only later declines which would be seen in older aSjD patients but not in young cSjD patients? Can the authors ascertain that this is not the case?

R#2: As for the “gradual” loss of Treg function in aSjD, we apologize that there are no publications available to directly support “gradual” loss” although it is commonly believed just based on general autoimmune etiology as the reviewer briefly commented. Since our current study is the first study to analyze cSjD immune cell subsets at the molecular and cellular level with the first Treg functional analysis, we would not know if cSjD “gradually” loses Treg function. As we are currently following up these patients on a regular basis, we will have better insights into the functional roles of Tregs in cSjD pathogenesis. We removed the word “Gradual” in the main body of the MS for this reason.

Regarding the question about reduced T reg function in aSjD, the controversy over the roles, numbers, and suppressive functions of Tregs in aSjD still exists. To our best knowledge, only one aSjD T reg functional assay study has been published, which indicates the impaired suppression of these cells (PMID: 19664141), where CD4⁺CD25⁺ and CD4⁺CD25⁻ cells at a 1:1 ratio in the mixed lymphocyte reaction were analyzed. Our current study on aSjD supports their result, while revealing that cSjD maintained the function in an *in vitro* analysis. The reference was added to the main body (Results and Discussion) of the revised MS (**yellow-highlighted**) (**Pages 10 and 16**)

We currently postulate that loss of Treg functionality in SjD might be associated with TOX expression in cSjD Treg. When we performed ShinyGO (Gene Ontology analysis) using cSjD-Treg-specific DEGs (**Supplementary Table 7 in the MS**), the predicted functions obtained from all available gene sets included “MiR-23-3p target gene” (Enrichment FDR of 0.000380665 and fold enrichment of 2.9). The Tox gene is a target gene of miR-23. Loss of miR-23 and ensuing TOX upregulation are associated with T-cell differentiation (PMID: 31844658). Downregulation of miR-23 decreases acetylation of FOXP3 of Treg, which appears to impair mitochondrial function via upregulation of sirtuin 1 and ROR γ t in Graves’ disease (PMID: 30391932). TOX⁺ Treg is more immunosuppressive by regulating chromatin accessibility (PMID: 31732165). Based on these studies, we hypothesize that TOX as an outcome of a highly regulatory circuit driven by miR-23 may compensate for Treg instability in the context of inflammation of cSjD patients. We plan to test this hypothesis in a chronological manner in the near future using single-cell ATAC-seq, with samples from our prospective cohort consisting of patients whose ages are anywhere between 4 to 90 years old.

The age information was added to **Supplementary Material (Pages 1-2)** and **Figure 2I figure legend (Page 28)**. In our study, a total of three pediatric cSjD patients (ages ranging from 12 to 18, newly diagnosed) samples and three adult SjD patients (ages ranging from 53 to 59, newly diagnosed or diagnosed for 1 or 5 years) samples with a total of six healthy controls (ages ranging from 15 to 54) were used for the Treg suppression assay. Two independent experiments for each child and adult group were performed.

Q#3-1. Were the Bx and BxRP conditions diagnosed as cSjD by a physician and if not, as what were they diagnosed since they show an overlapping set of symptoms with the patients classified as cSjD (Suppl. Fig. 1).

R#3-1. All cases were referred by physicians to the Center for glandular function evaluation and a lip biopsy. The physicians performed a comprehensive blood and clinical work-ups to rule out

other autoimmune conditions such as lupus, JIA, or juvenile parotitis prior to referrals. When they had a suspicion of cSjD based on the blood test or clinical evaluation, such as upregulated inflammatory markers, cytopenia, and/or SSA+, and fatigue, arthralgia, RP, sicca symptoms, syncope, or gastroparesis, respectively. Symptomatic non-SjD patients are being followed up every 6 months for a possibility of developing SjD-like conditions as they grow.

Q#3-2. What is the biopsy-negative non-cSjD coming from? Does this group mean that they did not have focal lymphocytic lesions, but just infiltration of inflammatory cells? Is this group another autoimmune disorders such as SLE?

R#3-2. The non-cSjD group in the study have neither focal lymphocytic infiltration nor infiltration of inflammatory cells. They simply have healthy glandular parenchyma. As the gold standard for cSjD for this study in the absence of pediatric specific diagnostic guideline, they are the ones who did not fulfil the 2016 criteria. Bx+non-cSjD patients have SjD-like focal infiltration, but they did not meet any other parameters in the 2016 criteria, failing to be diagnosed with cSjD. Therefore, patients are also followed up to monitor a possibility of presenting full-blown SjD phenotype or resolution of symptoms and signs as they grow. SLE was ruled out first for these cases prior to the referral for SjD evaluation.

Q#3-3. What is the meaning of having all the different groups if the main conclusions are based on identified differences between cSjD and HC? Throughout the manuscript the authors occasionally report of differences that are selectively present in one of the other groups but without explaining what it may mean.

R#3-3. Different group comparisons were explained in detail below (**Response #3-4**) and elaborated in the main body of MS in paragraphs with the addition of **Figure 1A and Supplementary Table 1A** for better presentation and clarification.

Q#3-4. Furthermore, how is the “non-cSjD” group different from healthy control?

R#3-4. As explained earlier, we used the 2016 ACR/EULAR criteria as the gold standard for diagnosis since there is no pediatric patient specific criteria available as of yet. The non-cSjD group is defined as the patients who visited UF Shands hospital for symptoms similar to SjD. However, they did not fulfill the 2016 ACR/EULAR criteria following comprehensive evaluations.

The 2016 criteria and experts’ opinions are what is being utilized most in the field of cSjD. The 2016 criteria include biopsy (3 points), serology (3 points), unstimulated salivary flow rate (1 point), occlusal surface staining (1 point), and Schirmer test (1 point). The summation of the points should be equal to or greater than 4 in order to fulfill the criteria for pSjD (PMID: 27785888). We had patients in our UF pediatric cohort with only biopsy positive (3 points) without meeting other criteria, only sialometry positive (1 point), or only serology positive (3 points), etc. All of these were considered to be non-cSjD patients.

Of these **non-cSjD patients**, we classified the patients **who are only biopsy positive (Bx+, 3 points)** as **Bx+ non-cSjD** so that we can identify what may be preceding the full-blown cSjD or SjD disease phenotype. Therefore, our non-cSjD group for this study exclude Bx+non-cSjD. The importance of **Bx+ non-cSjD** can be identified by the comparison between the transcriptomic

signatures of cSjD (all Bx+) and non-cSjD (Bx-). Also we can compare Bx+non-cSjD (Bx+) with non-cSjD (Bx-).

We also included **Bx and recurrent parotitis-positive patients (Bx+RP+ non-cSjD, 3 points)** because RP is the most common clinical presentation in children with SjD-like conditions, but is not included in the current aSjD criteria.

We also included cSjD (all Bx+) with two being RP+ and two being RP- to understand what contributes to RP. RP+cSjD and RP-cSjD can be compared in this group. This can also be achieved by comparing Bx+non-cSjD with Bx+RP+non-cSjD. Our rationale was that if any similar profile is identified from the two sets of comparisons just mentioned, that profile would be critical in driving RP. As RP is the most common reason for patient's clinic visits, we desired to identify transcriptomic evidence, its impact on RP, and impact of RP on cSjD so that RP (or salivary gland ultrasound) can be considered for future diagnostic criteria.

We presented the Bx+non-cSjD data set (**Response Figures 2, 3, Supplementary Figure 7J, 9B, 9C**), Bx-non-cSjD dataset (**Supplementary Figure 4C, 4E, 4F, 9A**), and RP data set (**Figure 5, Supplementary Figure 12**) in the main and supplementary MS, as requested.

In response to the comment by Reviewer #1 (Question #3-3) and Reviewer #2, we also performed **differential abundance analysis** (<https://rdrr.io/github/kstreet13/bioc2020trajectories/f/vignettes/workshopTrajectories.Rmd>),

which is to determine whether neighboring cells have the same condition or not and acquire better insights into key players unique in each group. As shown in **Response Figure 2** above, the imbalance score in the **Bx-negative non-cSjD (Panel F)** was similar to the imbalance score in **HC (Panel B)**.

Bx+ non-cSjD (red) shows a remarkable increase in the imbalance score in some **T, B, and monocyte subpopulations, compared to Bx-non-cSjD** (navy). Of note, Treg showed a dramatic increase in the imbalance score (Violin plot in **Response Figure 3**), potentially implying a compensatory mechanism against autoimmune activation in the Bx+non-cSjD and cSjD patients. These data indicate that differentially abundant cell populations may exist in the periphery of Bx+non-cSjD, driving the homing of immune cells to the target salivary glands.

Consistent with this result, predicted biological functions by the DEG-based GO analysis revealed that **effector CD8⁺ T cells from Bx+ non-cSjD groups up-enriched gene sets associated with effector function, including $\gamma\delta$ -T activation, compared to HC (Response Figure 4)**. Interestingly, the $\gamma\delta$ -T cell enrichment was not identified

in Bx-non-cSjD patients, **but in cSjD**, compared to HC. This may indicate that $\gamma\delta$ -T cell enrichment may be a common feature in the periphery of Bx+ individuals regardless of cSjD

diagnosis. We included this result in the **Supplementary Figure 7J and Discussion (yellow-highlighted) (Page 22)**.

For Bx-non-cSjD, M2-polarized monocytes highly expressed PLCG2, compared to Bx+ non-cSjD groups (Suppl Figure 4C and 4E). Indeed, PLCG2 is known to be important in determining myeloid lineage commitment and different functionality of monocytes (PMID: 24038146, 34157287). We postulate that **PLCG2** could be uniquely involved in **Bx-non-cSjD**, which requires further investigation, which was stated in **Discussion (yellow-highlighted) (Page 22)**.

Considering the minimal involvement of circulating immune cells in molecular interactions in Bx+ non-cSjD, we recently **performed a new scRNA-seq on the lip biopsy specimens** to examine the molecular events at the tissue level. Supporting our notion, the preliminary result revealed that tissue-infiltrating immune cells from Bx+non-cSjD are transcriptionally very different from circulating immune cells in the periphery. We performed integrated scRNA-seq analysis on 10 PBMC and 5 salivary gland-infiltrating immune cell samples from **Bx+non-cSjD** patients (**Response Figure 5 below**). Despite the small number of immune cells in the tissue, as we expected, tissue-infiltrating immune cells demonstrated strong enrichment of many inflammatory signature genes. This work is currently in progress.

Response Figure 5

Likewise, a similar trend was found in the integrated analysis of the tissue scRNA-seq data (**Response Figure 6** below) (5 PBMC and 3 salivary gland-infiltrating immune cell samples from **Bx-non-cSjD**).

Response Figure 6

A. Myeloid cells

B. B cells

C. CD4 T cells

D. CD8 T and NK cells

Taken together, to better define the clinical significance of Bx+ or Bx+RP+ non-cSjD groups, scRNA-seq analysis of tissue-infiltrating cells is required. However, considering the extensive data already presented in our current MS, please kindly understand that the primary goal of this study was to identify the transcriptomic profiles of cSjD compared to other pediatric groups for the first time in the field. Future studies involving our current tissue analysis and in vitro assays will further define the cSjD disease pathogenesis and transcriptomic profiles and the impact of RP+ and Bx+ on cSjD.

Q#4. Could it be more informative for a study that attempts to identify differences in cSjD versus aSjD to continually keep the cSjD patients with and without parotitis as separate groups and include in the study an internally generated aSjD dataset for direct comparison?

R#4: At the beginning of this project, we were fully aware that aSjD scRNA-seq data were already available by other groups (PMID: 36425764, 33603736, 36881472) and a large group of investigators collaborated to generate high-throughput data including scRNA-seq on aSjD. Therefore, the main focus of our study was to investigate the transcriptomic profiles of cSjD, identify the unique transcriptomic features of RP, and understand the significance of Bx+ without fulfilling the 2016 criteria in disease progression. And the secondary goal was to compare our data

with the published aSjD data and the high-throughput analysis on aSjD work in progress by other groups. For that reason, we did not include aSjD samples for scRNA-seq. Therefore, there are NO internally generated aSjD datasets available by our group at this present time.

When we analyzed our data for this MS, we realized that only one dataset (PMID: 33603736) out of three aSjD studies was accessible, which we included in our current MS. As many groups start publishing the data on aSjD from the current collaborative endeavor supported by NIH, we will certainly acquire their aSjD data to compare with our cSjD data in the future by acknowledging the value of the reviewer's comment. Alternatively, we can also include aSjD as we expand the scope and scale of our cSjD study.

Q#5. The authors are encouraged to improve the strength of their claims overall through the study by performing validation experiments for the expression of proteins that they consider to be functionally implicated in the mechanism of the disease.

R#5: To compare IFN gamma receptor-mediated STAT1 activation of CD4 T cells between cSjD and aSjD, we stimulated PBMCs derived from healthy controls (children and adults) and patients (cSjD and aSjD) (n=4/group) and stimulated them with 10 ng/ml of IFN-gamma (R&D system, cat#, 285-F-100) for 30 min, followed by flow cytometry analysis for phosphor STAT1 fluorescence (pSTAT1) (clone: KIKSI0803, fluorescence: EF660, Invitrogen, cat# 50-9008-42).

Interestingly, we found that IFN-treated CD4⁺ T cells from cSjD showed a robust increase in pSTAT1 expression upon stimulation, in which the magnitude of the increase was 269%, compared to child HC (**Response Figure 7, Panel A**). However, CD4⁺ T cells from aSjD only showed a low level

of pSTAT1 induction (33%) compared to adult HC. This additional analysis strongly supports that pronounced activation (or lower threshold for activation by IFN-gamma) of effector-like and memory CD4⁺ T cells in cSjD than in aSjD can be a key feature in cSjD pathogenesis. We included this cSjD-specific feature in **Figure 2J**.

In addition, CD8⁺ T cells showed the same trend, but the magnitude of pSTAT1 induction was not as robust as CD4⁺ T cells did (1.9% and 4.0% in adult SjD and cSjD, respectively) (**Response Figure 7, Panel B**). Similarly, IFN gamma did not significantly induce pSTAT1 expression in monocytes from cSjD and adult SjD, but the trend was still the same, showing a significantly higher level of pSTAT1 in cSjD than in adult SjD monocytes (**Response Figure 7, Panel C**).

Please kindly note that cSjD is a rare disease, which is underdiagnosed and understudied. Therefore, it is challenging to secure a sufficient amount of fresh blood samples for protein or functional analysis at this present time when only a vial of blood is being donated when they have

multiple vials drawn for a full panel of diagnostic work-ups. In the future, we will consider targeted recruitment rather than general recruitment to secure a sufficient volume of specimens for downstream functional applications.

Nonetheless, we additionally secured PBMC derived from HC and patients of children and adults to perform flow cytometry analysis of pSTAT1 expression following IFN-gamma stimulation of CD4⁺ T cells in response to the reviewer's comment. In addition, we further supported our findings by further analyzing published databases to rule out any influences of age or ethnicity on our monocyte datasets. We also searched protein databases, but unfortunately, there are no high-throughput protein data available, such as mass cytometry CyTOF. We sincerely appreciate Reviewer #1 for his/her invaluable comments.

Reviewer #2:

The presentation of results may need some focus, clarification and re-structuring.

--We sincerely thank the reviewer's comments.

Q#1. Definition of clinical cohorts is quite complex. I suggest adding a graphical representation of cohort definitions for easier readability. If possible, simpler names for the three non-cSjD cohorts might be introduced.

R#1: The graphic summary and a tabular summary to clarify 5 groups and their comparisons were added as **Figure 1A and Supplementary Table 1**, respectively. SjD is an internationally recommended abbreviation for Sjogren's disease. Bx is a general term to describe biopsy in practice, and RP refers to recurrent parotitis. We believe that the abbreviations used in this study are the simplest with the additional graphic and table summary provided. We also clarified these abbreviations where appropriate in the MS. The response #3 to the reviewer #1 also defined those groups and the rationale for the comparisons in detail.

Q#2. Line 116-122: Description of results from Suppl. Fig. 4 at this point in the MS seems odd and stops reading flow. M1/M2 polarization terminology has largely been abolished.

R#2: We agree with the reviewer. We added linking sentences to improve the reading flow for readers (**Lines 124-125 on Page 6**). In addition, please kindly note that wherever possible in the entire MS, we avoid using M1 or M2, but instead use "M1-like or M2-like". The gene set name in the figure was previously defined by the database.

Q#3. Figure 1: This figure is quite busy and could be streamlined. To illustrate description of results in the manuscript text, cluster presence in each clinical cohort should be shown (similar to Supplementary Fig 6E). Smaller data figures (Fig 1 H, I, J) could be moved to supplements instead. Fig 1L seems redundant, as data is presented in Fig 1M.

R#3: As the reviewer suggested, we transferred Figures 1H, 1I, and 1J to **Supplementary Figures 4H, 4I, and 4J**, respectively. We removed redundant genes (CCL4L2 and CCL3L1) in Figure 1M which makes Figure 1L unique. Please kindly note that we created a new **supplementary Figure 5** to transfer Supplementary Figures 4H, 4I, and 4J to **Supplementary Figures 5A, 5B, and 5C**, respectively. Accordingly, the Supplementary figure legend captions were also consecutively edited.

Q#4. Suppl. Fig. 2J offers interesting insights into relative cellular abundance in the different cohorts and could be presented as a main figure. In addition, I suggest differential neighborhood abundance testing to potentially identify previously unknown cellular players in different disease cohorts or in HC vs. cSjD(+/-non-cSjD) in a less supervised fashion.

R#4: Although we agree with the reviewer that **Fig. 2J** is an important finding, we believe that the population change data might be a better fit to remain in the supplementary data section because the main figure section currently focuses on immune cell subsets and sub-clusters.

In addition, we performed a differential neighborhood abundance test adopting an imbalance score(<https://rdrr.io/github/kstreet13/bioc2020trajectories/f/vignettes/workshopTrajectories.Rmd>).

Please kindly note that the imbalance scoring test did only provide UMAP, but not tSNE, thus we present a UMAP plot (**Response Figure 8, Panel A**) for convenience in which the cluster number is exactly the same as that of tSNE in our dataset.

For HC, the default imbalance score remains slightly increased in CD8⁺ T clusters (**Response Figure 8, Panel B**). More interestingly, in patients with cSjD, there was a remarkable increase in the imbalance score of almost all immune cell types especially in the score in CD8⁺ T and NK subpopulations (**Response Figure 8, Panel C**). It was consistent with our result in that cSjD clearly demonstrated disease positivity based on the data from GSEA, cell-cell interactions, and experimental validation. A similar pattern was observed in in Bx group (**Response Figure 8, Panel D**). Of note, there was a remarkable increase in the imbalance score in some T, B, and monocyte subpopulations, and Treg showed a dramatic increase in the imbalance score (**Response Figure 8, Panel G**), implying a compensatory mechanism against autoimmune activation in the Bx+ (red) and cSjD (sky blue) patients. In contrast, immune cells from non-cSjD (**Response Figure 8, Panel F**) did not have an increased imbalance score, generally showing a similar status or condition,

compared to HC. It was consistent with our results in that non-cSjD groups showed minimal and sporadic changes in disease-related molecules. Imbalance score from BxRP (**Response Figure 8, Panel E**) generally increased in CD4⁺ T subsets. These data support that many cellular players, e.g., CD8⁺ T cells, play a role in cSjD as well as Bx at the blood level.

Q#5. Suppl. Fig. 4: Datasets from adult SjD lack specific IFN-related monocytes - however, there is no direct DEG comparison with data from cSjD. This limitation should be discussed in the manuscript.

R#5: We agree with the comment. We revised the MS to include this as a limitation in **Discussion (yellow-highlighted) (Lines 416-417 on Page 17)**.

At the beginning of this project, we were fully aware that aSjD scRNA-seq data were already available by other groups (PMID: 36425764, 33603736, 36881472) and a large group of investigators collaborated to generate high-throughput data including scRNA-seq on aSjD. Therefore, the main focus of our study was to investigate the transcriptomic profiles of cSjD, identify the unique transcriptomic features of RP, and understand the significance of Bx+ without fulfilling the 2016 criteria in disease progression. And the second focus was to compare our data with the published aSjD data and the high-throughput analysis on aSjD work in progress by other groups. For that reason, we did not include aSjD samples for scRNA-seq. Therefore, there are no internally generated aSjD datasets available by our group at this present time.

When we analyzed our data for this MS, we realized that only one dataset (PMID: 33603736) out of three aSjD published studies was accessible, which we included in our current MS. As many groups start publishing the data on aSjD from the current collaborative endeavor supported by NIH, we will certainly acquire their aSjD data to compare with our cSjD data in the future by acknowledging the value of the reviewer's comment. Alternatively, we can also include aSjD as we expand the scope and scale of our cSjD study.

Q#6. Suppl. Fig. 6.: Results section to this supplement is quite long, text could be limited to most important findings.

R#6: We cut down **the Result for Suppl. Fig. 6.** significantly and edited it to only include important results.

Q#7. Figure 4: As the analysis method is only able to predict putative cell-cell interactions, the theoretical nature of presented cellular interaction data should be stated in the manuscript.

R#7: We agree that it is only an inference-based analysis using a previously established ligand-receptor interaction database. We have added the nature of the cell-cell interaction analysis to the discussion section, especially in the limitation section in **Discussion (yellow-highlighted) (Lines 555-557 on Page 22)**.

Q#8. Figure 4: Text of figure Panels C, D and E is too small and therefore illegible. A different way of presentation should be chosen.

R#8: We enlarged the size of Figures 4C and 4D to make it legible. In addition, due to space limitations, we moved Figures 4E and 4F to the new supplementary figures 9 and 11 and enlarged the figure size.

Q#9. Figure 4E: Bx and BxRP were grouped into one biopsy-positive group. Why was this grouping not performed for other analyses shown in Fig 1-3?

R#9: Analysis using biopsy positivity (Bx and BxRP combined) has been performed on all immune subsets. However, only meaningful results were presented in the MS overall due to the large volume of the data already presented in the MS.

Q#10. Line 825: no red circle is visible in Fig 5A.

R#10: The red circle in Figure 5A was enlarged to increase visualization.

Q#11: Fig 5G: see comment 8

R#11: We moved Figure 5G to the new supplementary Figure 12 and enlarged the size of the figure to make it legible. In addition, accordingly, the relevant captions of Figure 5 were reorganized accordingly.

Q#12: Figure 6 could be included in Fig 5.

R#12: We incorporated Figure 6 into Figure 5E.

Q#13. Line 369-371: Statement not suitable for data manuscript, should be rephrased. It was the last sentence in the second paragraph of the Discussion section.

R#13: We rephrased the sentence to make it suitable for the MS.

Revised in **Discussion** (Lines 386-388 on Page 16) to: Thus, future investigations of T cell receptor repertoires of effector CD4⁺ T cell subsets in the periphery and the target glandular tissue will provide insights into pathogenic T subsets in cSjD.

Q#14. Can the authors further comment on the criteria that were not met in non-cSjD and do the authors conclude that patients in the non-cSjD cohort present with a distinct disease or a combination of different diseases?

R#14: As explained earlier, we used the 2016 ACR/EULAR criteria as the gold standard for diagnosis since there is no pediatric patient specific criteria available as of yet. The non-cSjD group is defined as the patients who visited UF Shands hospital for symptoms similar to SjD. However, they did not fulfill the 2016 ACR/EULAR criteria following comprehensive evaluations.

The 2016 criteria and experts' opinions are what is being utilized most in the field of cSjD. The 2016 criteria include biopsy (3 points), serology (3 points), unstimulated salivary flow rate (1 point), occlusal surface staining (1 point), and Schirmer test (1 point). The summation of the points should be equal to or greater than 4 in order to fulfill the criteria for pSjD (PMID: 27785888). We had patients in our UF pediatric cohort with only biopsy positive (3 points) without meeting other criteria, only sialometry positive (1 point), or only serology positive (3 points), etc. All of these were considered to be non-cSjD patients.

Of these **non-cSjD patients**, we classified the patients **who are only biopsy positive (Bx+, 3 points)** as **Bx+ non-cSjD** so that we can identify what may be preceding the full-blown cSjD or SjD disease phenotype. Therefore, our non-cSjD group for this study exclude Bx+non-cSjD.

It is possible that the current non-cSjD may have a certain unidentified condition or a combination of different diseases, although SLE or JIA were ruled out prior to referrals, as they are symptomatic with varying degrees. We will certainly need the data at the tissue level and a long-term monitoring of these patients for a possibility of presenting a full-blown disease phenotype as they grow. Recently, we performed scRNA-seq using lip biopsy specimens (Our preliminary data are briefly presented in **Response Figures 5 and 6**) and we continue to monitor our patients in order to be able to answer the reviewer's question.

Q#15. Can the authors extrapolate from their data a specific molecule or migration receptor of blood leukocytes that is associated with RP and could be used diagnostically?

R#15: We proposed the mechanism and clinical relevance of RP-related genes of our interest in the **Discussion** (Lines 496-510 on Page 20) of the MS revision. The following genes are of our interest.

UBAP2 has been involved in various cellular processes, including protein trafficking and degradation of ubiquitinated protein. UBAP2 is reported to be responsible for UV-induced ubiquitylation of RNA polymerase II, preventing transcription of damaged RNA (PMID: 35633597). UBAP2 is predicted to be upregulated by subsets of microRNAs in the saliva of patients with aggressive periodontitis (PMID: 33124206). Aggressive periodontitis is associated with DNA damage (PMID: 24666368). Likewise, **CDC14A** is upregulated upon DNA damage (PMID: 26283732). Meanwhile, UBAP2 is highly upregulated in a subset of monocytes that have the potential to differentiate into osteoclast, suggesting the potential relation of UBAP2 in different lineage differentiation (PMID: 37339951). In our study, UBAP2 was significantly upregulated in the myeloid cells (especially, cluster 5 pDC) related to RP. Thus, we postulate that UBAP2 could be associated with nucleic acid damage and differentiation into another lineage, such as pDC. Alternatively, we postulate that cluster 5 pDC might harbor damage in nucleic acids within the nucleus and mitochondria in patients with RP in which the damaged DNA or RNA inside pDC might be initiating inflammatory cues mimicking bacterial or viral infections.

S100A8 and S100A12 are detectable in the salivary glands, serum, and saliva of individuals with SjD, involved in proinflammatory processes (PMID: 29998578, 31249499). The expression of S100A8 is downregulated during the maturation of peripheral blood monocytes to tissue macrophages (PMID: 3048809, 3313057, 3405210). **HLA genes, including HLA-DQB1**, are involved in presenting antigens to immune cells. In SjD, specific HLA gene variants, including HLA-DQB1, have been associated with an increased risk of developing the condition (PMID: 22001416, 3458307). KLF2 regulates proinflammatory activation of monocytes in chronic inflammatory conditions (PMID: 16617118). KLF2 also regulates apoptotic cell clearance in macrophages (PMID: 29150239). Thus, monocytes in RP condition are highly likely to be inflammatory. We consider that these activated monocytes could be involved in enhanced antigen presentation driven by monocyte-to-macrophage differentiation.”

We are currently testing these hypotheses to understand if either defective machinery of intracellular nucleic acid clearance in pDC or enhanced antigen presentation in monocytes is involved in patients with RP. The special focus of the latter was placed on mitochondrial double-

stranded RNAs accumulated in cSjD monocytes, which was recently funded by NIH/NIDCR (DE032707). Identifying and developing biomarkers for diagnostic, prognostic, and therapeutic interventions for RP in cSjD will be one of our ultimate goals in the grant.

Q#16. The presented data set is a valuable resource for the community and therefore should be made available via a public data database (if personal rights allow).

R#16: We have already deposited the data into the **NIAMS dbGaP database (Accession number: phs003048.v1.p1)** and made this statement in the data availability section in the MS.

Reviewers' comments:

Reviewer #1 (Remarks to the Author):

The authors have satisfactorily responded to my queries in all, but one matter. Namely, the claim that "A unique cluster of monocytes with type I and II IFN-related genes is highly specific to cSjD compared to adult SjD" (lines 58-59 of the abstract, and mentioned and discussed in lines 170-177 and 406-416) is not supported by reliable evidence. The authors very convincingly show that a cluster of monocytes with an IFN signature exists in a plethora of publicly available PBMC scRNAseq datasets as well as in their own, and that in cSjD the IFN signature of these cells is further elevated. Therefore, they prove that the existence of an IFN cluster is inherent to monocytes in PBMC samples in general. However, their comparison to aSjD is based on a singular dataset that is highly unusual in the sense that it does not contain an identifiable monocyte IFN cluster neither in its internal healthy nor diseased specimens. The authors say that they could not access other aSjD datasets (PMID: 36425764 and 36881472). However, authors' inability to analyse other datasets does not justify making a conclusion that is highly uncertain. It appears to this reviewer that a comparison to aSjD lies outside the scope of the current study, which even without it brings enough novelty as the first single-cell dissection of PBMCs in cSjD. Except the uncertainty of the comparisons made to aSjD, this study brings valuable and novel insights about childhood Sjögren's syndrome.

Reviewer #2 (Remarks to the Author):

All previous reviewer comments were addressed satisfactorily in the point-by point response letter and/or revised manuscript version. The current manuscript is an important, data-rich report on different groups of cSJD.

Minor comments:

- 1) The discussion section is more than double the length of a typical discussion section and contains a plethora of thoughts, conclusions, and literature. The section would profit from significant shortening by focusing on the most important findings of the manuscript and their relation to literature, as well as a brief outlook.
- 2) The fact that patients of the non-cSjD cohort are symptomatic to varying degrees and may thus have a unidentified condition (although SLE/JIA were ruled out) may be stated in the beginning of the manuscript.

AUTHORS' RESPONSE TO THE REVIEWERS' COMMENTS

We sincerely appreciate the reviewers' invaluable comments and suggestions (in purple), which certainly strengthened our manuscript (MS) during our second revision process. The summary of the comments and our responses (in black) are listed below.

Reviewer#1

Comment #1. Therefore, they prove that the existence of an IFN cluster is inherent to monocytes in PBMC samples in general. However, their comparison to aSjD is based on a singular dataset that is highly unusual in the sense that it does not contain an identifiable monocyte IFN cluster neither in its internal healthy nor diseased specimens. The authors say that they could not access other aSjD datasets (PMID: 36425764 and 36881472). However, authors' inability to analyze other datasets does not justify making a conclusion that is highly uncertain. It appears to this reviewer that a comparison to aSjD lies outside the scope of the current study, which even without it brings enough novelty as the first single-cell dissection of PBMCs in cSjD. Except the uncertainty of the comparisons made to aSjD, this study brings valuable and novel insights about childhood Sjögren's syndrome.

Response #1.

We thank the reviewer #1 for the comment. In agreement with the reviewer, we clarified the content related to the IFN-related monocyte cluster in cSjD by adding the statements below.

Lines 58-59: "A unique cluster of monocytes with type I and II IFN-related genes is identified in cSjD, compared to the age-matched control."

Line 171: " To compare cSjD monocytic clusters with aSjD monocytic clusters, we independently analyzed the published scRNA-seq dataset of aSjD PBMC, which is the only accessible aSjD dataset currently."

Line 179-180: " However, concluding whether it is due to cohort variability or true differences between cSjD and aSjD monocyte subsets will require further investigation."

Reviewer #2:

All previous reviewer comments were addressed satisfactorily in the point-by point response letter and/or revised manuscript version. The current manuscript is an important, data-rich report on different groups of cSjD.

Minor comments:

Comment #1) The discussion section is more than double the length of a typical discussion section and contains a plethora of thoughts, conclusions, and literature. The section would profit from significant shortening by focusing on the most important findings of the manuscript and their relation to literature, as well as a brief outlook.

Response #1: We significantly condense the lengthy discussion, leaving the most important findings in the MS with the content reorganization.

Comment #2) The fact that patients of the non-cSjD cohort are symptomatic to varying degrees and may thus have an unidentified condition (although SLE/JIA were ruled out) may be stated in the beginning of the manuscript.

Response #2: Thank you for your suggestion. We included the statement in the introduction section (Lines 103 and 532) in this second revision.

Line 103: “ symptomatic patients who failed to meet the 2016 SjD criteria”

Lines 532-533: “Finally, we presume that biopsy-negative non-cSjD (symptomatic) may have an unidentified autoimmune condition, which is to be further elucidated.”